# Scalable anisotropic cooling aerogels by additive freeze-casting

Kit-Ying Chan[1,2], Xi Shen ●[1,2] ✉, Jie Yang[1], Keng-Te Lin[3], Harun Venkatesan[1], Eunyoung Kim[1], Heng Zhang[1], Jeng-Hun Lee[1], Jinhong Yu[4], Jinglei Yang ●[1] & Jang-Kyo Kim ●[1,5] ✉

Cooling in buildings is vital to human well-being but inevitability consumes significant energy, adding pressure on achieving carbon neutrality. Thermally superinsulating aerogels are promising to isolate the heat for more energy-efficient cooling. However, most aerogels tend to absorb the sunlight for unwanted solar heat gain, and it is challenging to scale up the aerogel fabrication while maintaining consistent properties. Herein, we develop a thermally insulating, solar-reflective anisotropic cooling aerogel panel containing in-plane aligned pores with engineered pore walls using boron nitride nanosheets by an additive freeze-casting technique. The additive freeze-casting offers highly controllable and cumulative freezing dynamics for fabricating decimeter-scale aerogel panels with consistent in-plane pore alignments. The unique anisotropic thermo-optical properties of the nanosheets combined with in-plane pore channels enable the anisotropic cooling aerogel to deliver an ultralow out-of-plane thermal conductivity of 16.9 mW m$^{-1}$ K$^{-1}$ and a high solar reflectance of 97%. The excellent dual functionalities allow the anisotropic cooling aerogel to minimize both parasitic and solar heat gains when used as cooling panels under direct sunlight, achieving an up to 7 °C lower interior temperature than commercial silica aerogels. This work offers a new paradigm for the bottom-up fabrication of scalable anisotropic aerogels towards practical energy-efficient cooling applications.

The global warming due to climate change leads to a surge in the use of air conditioning in many regions of the world over the past decades. The energy consumption required for space cooling in many sectors, such as buildings, electric vehicles and food supply chains, is one of the major contributors to carbon dioxide emissions, seriously impeding the global efforts towards carbon neutrality[1]. To achieve energy-efficient cooling, thermal insulation materials with an ultralow thermal conductivity ($k$) are essential to reduce the parasitic heat gain by retarding the heat transfer from hot ambient to cool interiors. Conventional insulation materials, such as polyurethane (PU), expanded polystyrene (EPS) and extruded polystyrene, are commonly used because of their low costs and low $k$ in the range of 30–40 mW m$^{-1}$ K$^{-1}$ [2]. To further inhibit the heat transfer, state-of-the-art thermal insulators, like gas filled panels, phase change materials and aerogels, have been developed, showing suppressed $k$ of less than that of the air (24 mW m$^{-1}$ K$^{-1}$). In addition to the ultralow $k$, their use in outdoors also demands a low solar absorption to mitigate the solar heat gain under direct sunlight. However, it is challenging to achieve both an ultralow $k$ and a high solar reflectance,

[1]Department of Mechanical and Aerospace Engineering, The Hong Kong University of Science and Technology, Clear Water Bay, Kowloon, Hong Kong. [2]Department of Aeronautical and Aviation Engineering, The Hong Kong Polytechnic University, Hung Hom, Kowloon, Hong Kong. [3]Centre for Translational Atomaterials, Swinburne University of Technology, Hawthorn, Melbourne, VIC 3122, Australia. [4]Key Laboratory of Marine Materials and Related Technologies, Ningbo Institute of Materials Technology and Engineering, Chinese Academy of Sciences, Ningbo 315201, China. [5]School of Mechanical and Manufacturing Engineering, University of New South Wales, Sydney, NSW 2052, Australia. ✉e-mail: xi.shen@polyu.edu.hk; mejkkim@ust.hk

necessitating additional laborious processes to apply solar-reflective paints on the surface of thermal insulation materials.

Aerogels are highly porous materials made from nanostructured constituents for a broad range of applications, including electromagnetic interference (EMI) absorption[3], energy storage[4–6], and environmental remediation[7]. Thanks to their ultralow densities and high porosities, aerogels made from polymers[8,9], ceramics[10], and carbon nanomaterials[11] are promising alternatives to conventional thermal insulation materials. In particular, anisotropic aerogels containing highly aligned pores have been considered more effective than their isotropic counterparts for thermal insulation[12,13]. The anisotropic structures yield anisotropic $k$ in the two orthogonal directions, reducing the heat transfer transverse to pore alignments and thus achieving thermal superinsulation[13–18]. The $k$ values measured in the direction transverse to the aligned pores, $k_{trans}$, were typically 15–37 mW m$^{-1}$ K$^{-1}$, much lower than those in the pore alignment direction ($k_{align}$ = 44–170 mW m$^{-1}$ K$^{-1}$). A high anisotropic factor, $k_{align}/k_{trans}$, of over 10 was attained by introducing graphene oxide (GO) sheets in the aligned nanocellulose cell walls, leading to a greatly suppressed $k_{trans}$ of 15 mW m$^{-1}$ K$^{-1}$ [14]. The high $k_{align}$ allowed the dissipation of incoming heat along the alignment direction, which in turn reduced the heat flow in the transverse direction. Such anisotropic aerogels with very high anisotropic factors are particularly useful for building envelopes where thermal insulation is only required in one direction between the interior and exterior environments to mitigate their heat exchanges, enabling to use thinner materials for insulation compared to conventional isotropic foams[19]. Nevertheless, there still lacks a clear strategy to tailor the anisotropic factor such that optimal thermal insulation in the transverse direction can be achieved. Moreover, most of the aerogels, especially carbon-based ones, tended to absorb the solar radiation rather than reflecting it despite their low $k$ values[20]. Although several attempts have been made to integrate low $k$ with high solar reflection in polymer films[21] and aerogels[22] by tailoring their microstructures, their $k$ values remained isotropic. Therefore, it is highly desirable to develop an anisotropic aerogel having in-plane pore channels with highly anisotropic $k$ and excellent solar reflectance for more energy-efficient cooling.

Despite the many available techniques to fabricate anisotropic aerogels, it remains a great challenge to achieve in-plane pore channels with consistent sizes and alignments in an aerogel panel having large lateral dimensions in the decimeter scale. A porous nanowood having in-plane pore channels of over 10 cm in length was developed by removing lignin from the natural wood[23], but the top-down approach employed therein was rarely extended to materials other than nanocellulose because of raw material limitations. By contrast, freeze-casting is an effective bottom-up method to construct aerogels having aligned pores using various nanoscale constituents. The directional freeze-casting technique has been widely used to prepare anisotropic aerogels with aligned pore channels[24–27]. Compared to other methods, the freeze-casting is simple, environmentally friendly, and inexpensive[28]. More importantly, the pore morphology and pore size of the aerogels can be easily tailored by adjusting the composition of colloidal solution and freeze-casting parameters, such as cooling rate and temperature gradient[29,30]. Organic[31,32], ceramic[33–35], and carbon aerogels[14,17,36,37] having tubular or lamellar pores were designed from various polymers, nanofibers, and nanosheets using uni- or bi-directional freeze-casting. However, the scaling up of freeze-casting to achieve decimeter-long pore channels with consistent alignments and sizes is tough because of the difficulty in maintaining the low temperature required for the directional ice growth when the solidification front advances away from the cold source. In fact, it was reported that the pore density differed by ~30% at two different positions with only ~1 cm apart along the direction of ice growth from a cold source[14]. Although uniform pore alignments over a longer freezing distance of 3 cm were

realized using double-side cold sources[38], such a lateral dimension is still too small for practical applications.

Herein, we designed an anisotropic cooling aerogel (ACA) panel containing in-plane aligned pores with engineered pore walls to deliver both anisotropic $k$ and excellent solar reflectance (Fig. 1a). Inspired by the continuous layer-by-layer process of additive manufacturing[39], an additive freeze-casting technique was developed for fabricating decimeter-scale, anisotropic aerogel panels (Fig. 1b). The additive freeze-casting allowed the cumulative freezing of colloidal solution from one side in a block-by-block manner while maintaining a short freezing distance in each frozen block, resulting in aerogel panels having in-plane pore channels with consistent alignments and pore sizes across the entire freezing distance (Fig. 1c and Supplementary Fig. 1). Highly flexible, anisotropic aerogel panels with various dimensions and thicknesses were demonstrated using the versatile additive freeze-casting technique (Fig. 1d). Further engineering the aligned pore walls by incorporating two-dimensional (2D) boron nitride nanosheets (BNNS) in waterborne polyurethane (WPU) gave rise to an ACA with highly anisotropic $k$ and high solar reflectance by taking advantage of the unique anisotropic thermo-optical properties of BNNS. Using the additive freeze-casting, a large ACA panel of 20 cm × 20 cm in lateral dimensions were made from the BNNS/WPU colloidal solution, exhibiting an ultralow $k$ (≤24 mW m$^{-1}$ K$^{-1}$) and a high solar reflectance (≥90%) consistently throughout the whole panel. The real-world cooling performance of the ACA panel under direct sunlight was superior to commercial insulation materials. This work offers a bottom-up strategy for the facile and scalable fabrication of anisotropic aerogels with consistent pore alignments for energy-efficient cooling applications.

## Results

### Design and additive freeze-casting of large-scale anisotropic aerogels

We designed the ACA panel with in-plane aligned pores (Fig. 1a) aiming to achieve the following attributes simultaneously, namely, (i) high anisotropy with an ultralow $k$ in the thickness direction, (ii) high optical reflectance in the solar wavelength ranging 0.3–2.5 μm, (iii) consistent thermal and optical performance throughout the panel of decimeter scale, (iv) mechanically flexible and robust to be rolled up for easy storage and transportation, and (v) cost-effective materials and processing. As such, WPU was chosen as the main constituent of the pore walls because PU foams are a common insulation material with good mechanical flexibility[40,41]. The typical white appearance of PU foams also testifies to their good visible light reflection. To further reduce the $k$ in the thickness direction and improve the solar reflectance, BNNS were introduced in the aligned WPU walls to take advantage of the unique anisotropic thermo-optical properties of 2D BNNS. The high in-plane $k$ of BNNS can redirect the heat flow in the pore alignment direction, in turn inhibiting the heat transfer through the thickness to reduce the parasitic heat gain (Fig. 1a)[42]. Furthermore, $h$-BN has a large band gap of 5.765 eV[43], corresponding to the absorption of light with wavelengths up to approximately 215 nm. This value is even smaller than the lower bound of solar spectrum (~300 nm), meaning that $h$-BN is intrinsically unable to absorb sunlight and thus favorable for a low solar heat gain[44].

To enable the scale-up fabrication of the ACA panel with uniformly aligned pores in the panel plane, an additive freeze-casting technique was developed, as shown in Fig. 1b. Supplementary Movie 1 displays the additive freeze-casting of decimeter-scale ACA panel in action. Unlike conventional freeze-casting with a fixed cold source, a moving cold source equipped with a solution dispenser was used in the casting process which offered high flexibility in freezing distance for achieving consistent pore alignments. Moreover, with continuous supply of liquid nitrogen to the moving cold source, consecutive directional freezing of the solution in the gap between

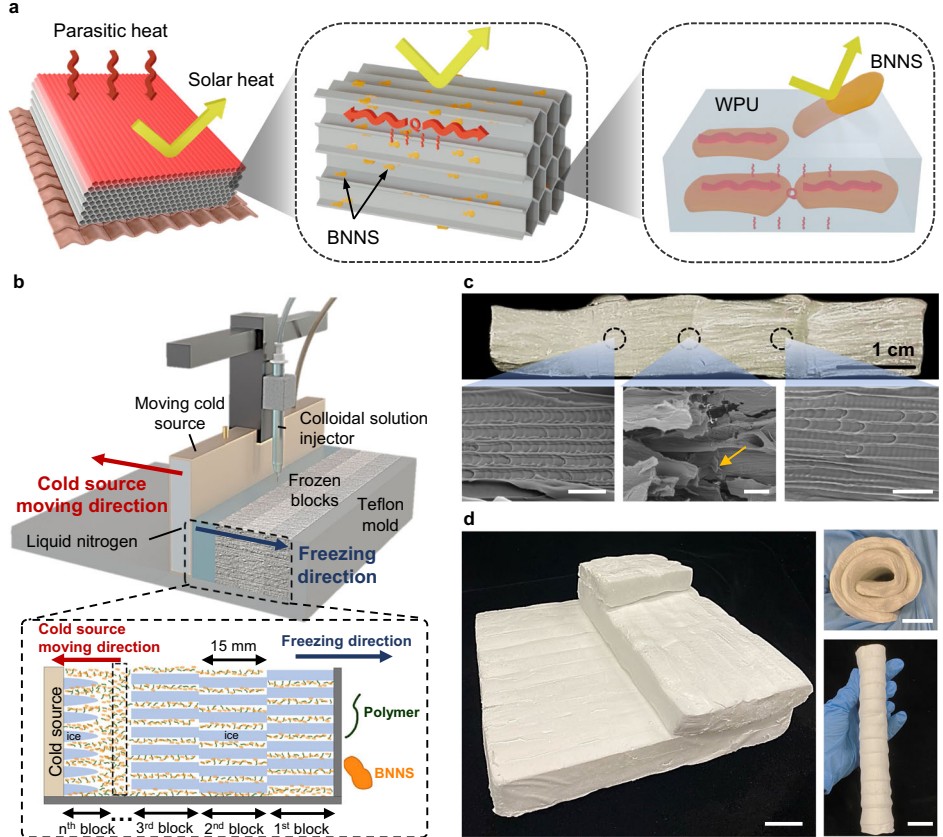

**Fig. 1 | Design and additive freeze-casting of decimeter-scale anisotropic aerogels. a** Schematics of the multiscale structures responsible for ultralow $k$ and high solar reflectance of the anisotopic aerogel panel. **b** Schematics of the set-up and operating principle of the additive freeze-casting for the fabrication of decimeter-scale, anisotropic aerogel from BNNS/WPU colloidal solution. **c** Photograph and scanning electron microscope (SEM) images showing the uniform pore alignment and structural integrity of the aerogel. The SEM images were taken from the different spots in the pore alignment direction, showing the uniform pore diameters along the freezing direction and the seamless connection between adjacent blocks as indicated by an orange arrow. Scale bars: 50 μm. **d** Photographs of anisotropic aerogel panels of different dimensions fabricated using the additive freeze-casting. They are highly flexible to be rolled up for easy storage and transportation. Scale bars: 2 cm.

the cold source and another end, i.e., either the Teflon mold (i.e., 1st block) or the frozen block, was possible until a desired dimension of aerogel panel was realized (see Methods for details). Therefore, our additive freeze-casting approach combined the continuous process of additive manufacturing and controlled pore morphology of directional freeze-casting, cumulatively freezing the colloidal solution in a block-by-block manner while maintaining consistent alignment in each frozen block (Fig. 1b). To demonstrate the high-quality alignment, we fabricated a seamlessly connected aerogel panel with a lateral dimension of 5 cm (i.e., four blocks in the freezing direction) by 5 cm (transverse to the freezing direction) and a thickness of 1 cm (Fig. 1c and Supplementary Fig. 1). The SEM images taken from different frozen blocks show in-plane aligned pore channels (Fig. 1c and Supplementary Fig. 1), in contrast to the randomly aligned pores in the aerogel obtained by conventional unidirectional freeze-casting (Supplementary Fig. 2). The pore sizes were measured from the cross-sectional SEM images taken at different locations along the freezing direction, indicating highly consistent pore sizes of ~30 μm along the entire dimension (Supplementary Fig. 3). Such consistent pore morphologies were essential to uniform thermal and optical properties across the whole aerogel panel (Fig. 1a). Furthermore, the cell walls of adjacent frozen blocks were well connected at the boundaries (orange arrows in Fig. 1c) owing to the recrystallization of ice[45]. This suggests that the structural integrity of the aerogels fabricated using the additive freeze-casting was not compromised despite the presence of block boundaries. Aerogels with different dimensions were fabricated using the additive freeze-casting

technique, as shown in Fig. 1d. The aerogel panels exhibited excellent flexibility for roll-up storage and transportation (Fig. 1d and Supplementary Movie 2), confirming their flexibility and structural integrity.

## Optimization of microstructures of ACA

The composite aerogel made from BNNS and WPU had an ultralow density and white appearance with highly anisotropic microstructures, featuring porous cross-sections and aligned cell walls in the freezing direction (Fig. 2). The Fourier-transform infrared microscopy (FTIR) spectrum of composite aerogel showed a new peak in addition to that of WPU[46] at ~805 cm$^{-1}$ (Fig. 2b), corresponding to the in-plane B-N-B bond stretching and thus confirming the presence of BNNS. To concomitantly achieve an ultralow density, a high porosity, and aligned cell walls essential for an anisotropic $k$, the concentrations of WPU and BNNS in the colloidal solution were further optimized, resulting in different physical properties and microstructures of the BNNS/WPU composite aerogels (Fig. 2c, d). The pristine WPU aerogel made from a high concentration WPU solution of 10 wt% showed thick pore walls with a high density of ~110 mg cm$^{-3}$ and a low porosity of 89.1%. When the WPU concentration was reduced from 10 wt% to 1.4 wt%, the density significantly dropped to 23.9 mg cm$^{-3}$ while the porosity rose to 97.7%, showing the two properties being inversely proportional with respect to WPU concentration (Supplementary Fig. 4). Nevertheless, the cell walls became thinner and less regularly aligned due to the low solid concentration (Fig. 2d and Supplementary Figs. 5, 6), which was detrimental to the mechanical integrity of the WPU aerogel and

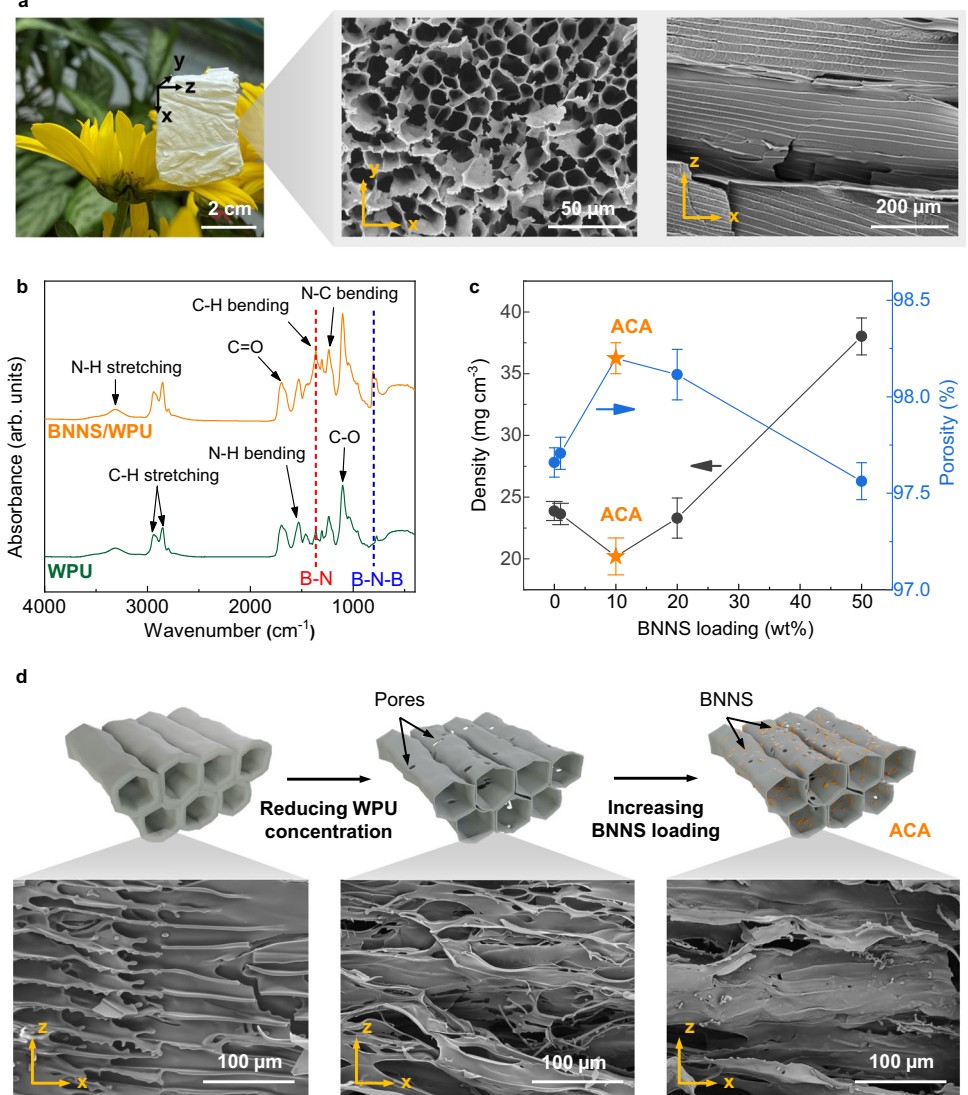

**Fig. 2 | Microstructures and physical properties of composite aerogels.**
**a** Photograph of an ultralight areogel and the corresponding SEM images showing the highly porous cross-section and aligned pore walls. **b** FTIR spectra of pristine WPU and BNNS/WPU composite aerogels. **c** Densities and porosities of BNNS/WPU aerogels at different BNNS loadings. Error bars represent standard deviations. The ACA showed the lowest density and highest porosity. **d** Schematics and SEM images showing the effects of WPU concentration and BNNS loading on the microstructures of pore walls.

generated significant shrinkage of almost 50% during freeze-drying (Supplementary Fig. 7). To mitigate the shrinkage and improve the alignment, BNNS were added at different loadings (i.e., mass percentage of BNNS in the entire BNNS/WPU mixture) in the 1.4 wt% WPU solution. Interestingly, the density of BNNS/WPU further dropped to 20.2 mg cm$^{-3}$ when the BNNS loading was raised to 10 wt% despite the higher density of BNNS (2.25 g cm$^{-3}$) compared to that of WPU (1.02 g cm$^{-3}$) (Fig. 2c), leading to a concurrently augmented porosity of 98.2%. This anomaly occurred because the rigid BNNS fillers functioned as effective reinforcement of pore walls[47], which in turn reduced the shrinkage and gave rise to an even lower density of composite aerogel than the pristine WPU aerogel counterpart. Moreover, improved alignments in the composite aerogel were also possible thanks to the high aspect ratio of 2D BNNS[48]. A further increase in BNNS loading to 50 wt% reduced the voids in the pore walls (Supplementary Fig. 6), yet also caused an undesirable surge in density arising from the high density of BNNS. To summarize, a low density of 20.2 mg cm$^{-3}$, a high porosity of 98.2%, and highly aligned BNNS/WPU pore walls were realized at an optimal BNNS loading of 10 wt% and WPU concentration of 1.4 wt%, which were used to fabricate the ACA panel. These

structural features ultimately translated into anisotropic $k$ and excellent solar reflectance with much diminished heat gains for more energy-efficient cooling.

## Anisotropic thermal conductivities of ACA

To achieve thermal superinsulation in the thickness direction of the ACA, an ultralow $k$ in the transverse to alignment direction, $k_{trans}$, is essential. Moreover, for anisotropic aerogels, a concurrently high $k$ in the alignment direction, $k_{align}$, is highly desirable to yield a high anisotropic factor, $R = k_{align} / k_{trans}$, further reducing the heat transfer in the transverse direction by diverting the heat flow in the alignment direction[49]. The unique microstructural features of the ACA are conducive to both an ultralow $k_{trans}$ and a high $R$. First, simultaneously a low density and high porosity of ACA (Fig. 2c) was the prerequisite for an ultralow $k_{trans}$. As shown in Supplementary Fig. 8, the $k_{trans}$ reduced with decreasing WPU concentration, manifesting a strong correlation with the density. Furthermore, by exploiting the high in-plane $k$ of BNNS in the aligned pore walls, higher anisotropic $k$ values were attained proportional to BNNS loading. As shown in Fig. 3a, $k_{align}$ improved by ~120% when the BNNS loading was increased to 50 wt%

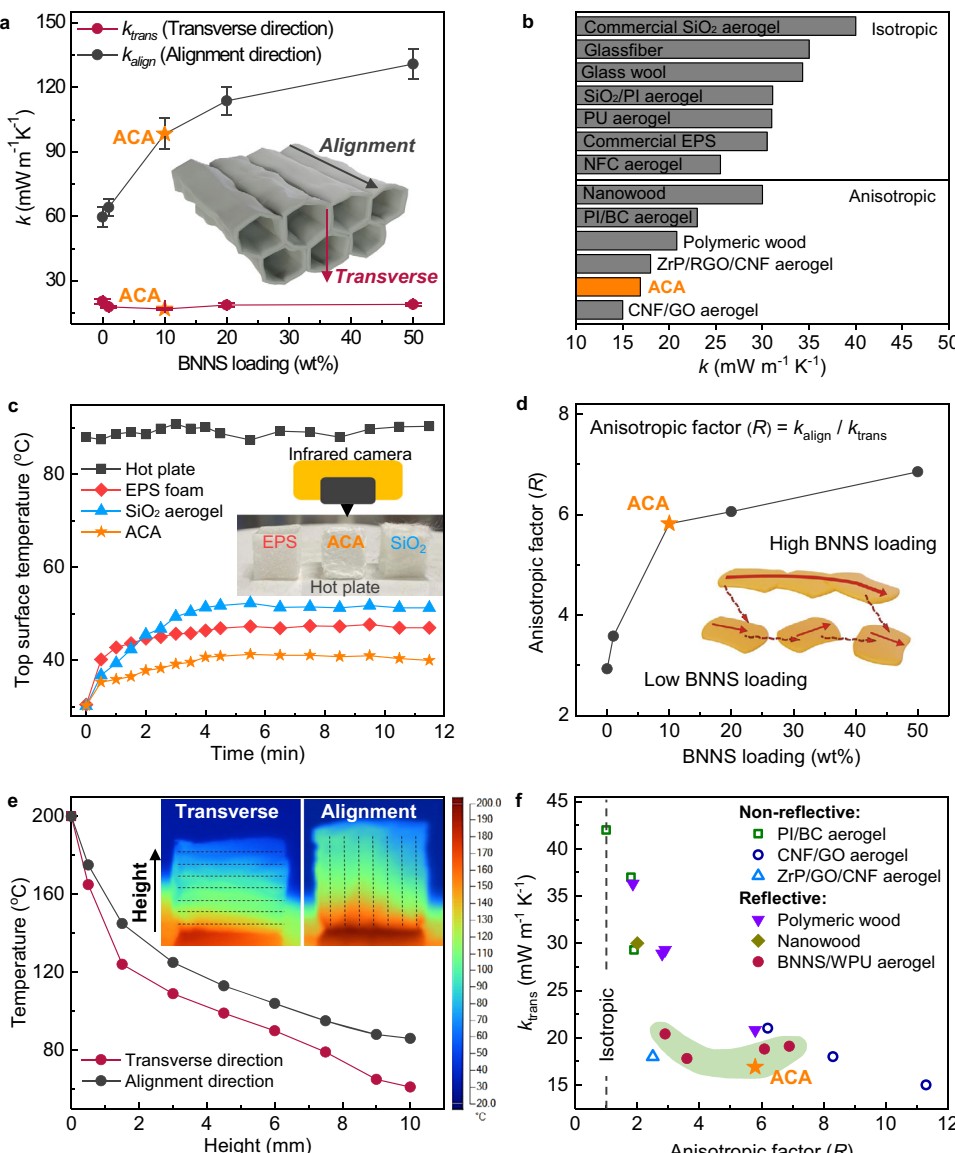

**Fig. 3 | Anisotropic thermal conduction in ACA. a** Effect of BNNS loading on thermal conductivities of aerogels in two directions. Error bars represent standard deviations. **b** Comparison of thermal conductivities of ACA with other thermal insulating materials available commercially and reported in the literature, including both isotropic materials (Commercial SiO$_2$ aerogel, fiberglass[53], glass wool[54], silica/polyimide (SiO$_2$/PI) nanocomposite aerogel[50], PU aerogel[40], commercial EPS foam and nanofibrilated cellulose (NFC) aerogel[51]) and anisotropic materials (nanowood[23], PI/bacterial cellulose (BC) aerogel[15], polymeric wood[13], zirconium

phosphate (ZrP)/reduced GO (RGO)/CNF aerogel[52], and CNF/GO aerogel[14]). **c** Evolution of top surface temperature of the ACA measured using thermal camera in comparison with commercial EPS foam ($k = 30.5$ mW m$^{-1}$ K$^{-1}$) and SiO$_2$ aerogel ($k = 40.0$ mW m$^{-1}$ K$^{-1}$). **d** Effect of BNNS loading on anisotropic factor ($R$) of composite aerogels. **e** Temperature distributions and infrared images showing the heat conduction along both alignment and transverse directions. The dash lines indicate the alignment of pore walls. **f** Comparison of $k_{trans}$ and $R$ of ACA with different anisotropic materials reported in the literature[13–15,23,52].

aided by highly aligned BNNS[42]. By contrast, the $k_{trans}$ was almost insensitive to the addition of BNNS, a reflection of the much lower out-of-plane $k$ of BNNS than the in-plane direction counterpart[42]. In fact, the $k_{trans}$ even marginally declined when the BNNS loading was increased to 10 wt%, a functionally similar trend of density (Fig. 2c), thus giving rise to an ultralow value of 16.9 mW m$^{-1}$ K$^{-1}$. It is worth mentioning that this value is even lower than the vast majority of insulating materials available commercially and reported in the literature (Fig. 3b)[13–15,40,50–54], indicating excellent thermal insulation performance of the ACA. Although an anisotropic GO-based aerogel exhibited a slightly lower thermal conductivity of 15 mW m$^{-1}$ K$^{-1}$ than our ACA[14], the former aerogel failed to effectively reflect the sunlight due to the intrinsically high optical absorption of GO in the solar spectrum[55]. The thermal insulation performance of the ACA was also confirmed by its thermal response of top surface when heated on the bottom using a

hot plate at 90 °C, as shown in Fig. 3c. The top-surface temperature of ACA gradually rose to a steady-state temperature of only ~41 °C, equivalent to 49 °C lower than the bottom surface. In comparison, the top surface of commercial EPS foam ($k = 30.5$ mW m$^{-1}$ K$^{-1}$) and SiO$_2$ aerogel ($k = 40.0$ mW m$^{-1}$ K$^{-1}$) became 6 and 10 °C hotter than that of ACA under the same condition, respectively, substantiating better thermal insulation performance of the ACA.

In addition to the ultralow $k_{trans}$, a high $R$ was also achieved in the ACA thanks to the high $k_{align}$ arising from the aligned BNNS. As shown in Fig. 3d, $R$ increased sharply with raising BNNS content initially and reached ~6 for ACA. Further increasing BNNS content only slightly improved the $R$ to ~7 at 50 wt% at the expense of a higher $k_{trans}$ than ACA (Fig. 3a). The anisotropic heat transfer was clearly visualized by monitoring the heat flow along the two orthogonal directions using a hot plate at 200 °C as the heat source, as shown in Fig. 3e. At a distance

of 10 mm from the heat source, the average temperature reached 86 °C in the alignment direction after 60 s. By contrast, the average temperature was only 61 °C in the transverse direction under the same condition, substantiating better insulation performance in the transverse than alignment directions. This finding has a practical implication in which a higher $R$ of the anisotropic material would facilitate better heat dissipation in the horizontal plane, significantly reducing its thickness required to achieve the same insulation performance as the isotropic material with a similar $k$[19]. Moreover, a high $R$ avoided local heat concentrations which may degrade polymers, benefiting long-term performance of the ACA. Given the importance of both $k_{trans}$ and $R$ for anisotropic insulating materials, these properties of the ACA are compared with other anisotropic materials to assess its relative insulation performance, as shown in Fig. 3f. While a GO-based aerogel showed a much higher $R$ of 11.3 than our ACA, its $k_{trans}$ (15 mW m$^{-1}$ K$^{-1}$) was only ~11% lower than that of ACA (16.9 mW m$^{-1}$ K$^{-1}$). Worse yet, the GO-based aerogel cannot be used for effective thermal insulation under sunlight because of the intrinsic solar absorption characteristic of GO. Among solar-reflective insulating materials, our ACA delivered both a higher $R$ and lower $k_{trans}$ than nanowood[23], while it also exhibited a lower $k_{trans}$ than polymeric wood having the same $R$ value of 5.8[13]. These findings highlight the effectiveness of using high-$k$ BNNS to achieve a high $R$ without adversely affecting $k_{trans}$ by tailoring BNNS content in highly aligned cell walls so that the efficiency of directional insulation would be realized. Similar to the heat conduction, the mechanical properties of ACA were also anisotropic due to the anisotropic structure (Supplementary Fig. 9). Both the compressive modulus and strength of aerogels increased with increasing BNNS loading as a reinforcement effect of BNNS[56] with strengths of 3.71 and 7.37 kPa in the transverse and alignment directions, respectively. These values compare well with those of common aerogels, such as covalently crosslinked polymer aerogels[31], graphene aerogels[57], and MXene aerogels[58,59].

## Solar reflectance of ACA

Apart from thermal superinsulation, our ACA also demonstrated excellent optical reflectance towards solar irradiation, reducing the heat gain from sunlight for more effective daytime thermal insulation. As shown in Fig. 4a, the commercial EPS foam and SiO$_2$ aerogel showed relatively low reflectances of ~80% across the whole solar spectrum despite their good thermal insulation. This means that ~20% of the incident sunlight was absorbed by these porous materials given the negligible transmission (Supplementary Fig. 10), raising the surface temperatures and compromising their thermal insulation properties under sunlight. Coating a commercial high-reflective paint on the SiO$_2$ aerogel surface ameliorated the reflectance in the visible (VIS, 0.4 to 0.8 μm) and near infrared A (NIR-A, 0.8 to 1.4 μm) wavelengths to ~90%, but was inefficient in the NIR-B region (1.4 to 2.5 μm) because of the intrinsic IR absorption of organic molecules of the paint[60]. By contrast, the ACA reflected nearly 100% incident irradiation in the VIS-NIR-A wavelengths, which account for ~90% of the total energy emitted from the sun, with a better reflectance in the NIR-B region than the commercial paint. Consequently, the solar-weighted reflectance of the ACA reached 97 % across the whole solar wavelengths, which was 23%, 20% and 14% higher than the commercial EPS foam (79%), SiO$_2$ aerogel (81%) and coated SiO$_2$ aerogel (85%), respectively. Such an excellent reflectance makes the ACA an intriguing candidate for reducing the solar heat gain without an additional reflective coating.

To understand the mechanisms behind the excellent sunlight reflection of the ACA, the effects of porosity and BNNS addition on reflectance were probed by both experiments and the finite-difference time-domain (FDTD) simulations (see details in Methods). Experimentally, the porosity of the final aerogel was found inversely proportional to the WPU concentration (Supplementary Fig. 4). The solar reflectance increased from 8 to 94% when the WPU structure was

changed from a solid film (with a 0% porosity) to an aerogel at a WPU concentration of 1.4 wt% (with porosity of 97.7%) (Supplementary Fig. 11). The same aerogel also delivered the lowest $k_{trans}$ value (Supplementary Fig. 8), indicating potential synergy arising from the optimal WPU concentration for desired solar reflectance and thermal conductivity. The benefit of high porosity was further confirmed by the FDTD simulations (Fig. 4b and Supplementary Fig. 12). As shown in Supplementary Fig. 12a, the porosity of simulation models was controlled by varying the inter-pore distance with a fixed pore diameter of 20 μm according to the experimental observations (Supplementary Fig. 1). The simulated reflection spectra for different porosities are compared in Supplementary Fig. 12b with the resulting reflectance values shown in Fig. 4b. A higher porosity led to a monotonic rise in simulated reflectance, consistent with the experimental trend (Supplementary Fig. 11a). Furthermore, the reflectance showed a direct correlation with the solid/air interfacial area (Supplementary Fig. 12c). This means that the large interfacial area arising from the high porosity of ACA offered extensive solid/air interfaces for multiple scattering due to the mismatched refractive indices of air and solid, contributing to the high reflectance of ACA.

The effect of BNNS addition on reflectance of ACA was also investigated using the FDTD simulations. The presence of BNNS fillers enhanced the reflectance of the aerogel with a high porosity by 22% against that without BNNS, as shown in Fig. 4c, proving the positive role of BNNS in realizing the high reflectance of ACA. The experimentally measured solar-weighted reflectance initially surged to 97% when the BNNS loading was increased to 10 wt%, followed by a gradual decline with further increasing the BNNS loading (Supplementary Fig. 13a). Of note is that such an optimal BNNS loading for the high reflectance also led to the lowest thermal conductivity in the transverse direction (Fig. 3a), both of which resulted from the highest porosity achieved at a BNNS loading of 10 wt% (Fig. 2c). Combining an ultralow $k$ of 16.9 mW m$^{-1}$ K$^{-1}$ in the thickness direction and a high reflectance of 97%, the ACA outperformed existing state-of-the-art insulation materials[21–23,61–66], none of which possessed both a $k$ lower than that of air (24 mW m$^{-1}$ K$^{-1}$) and a reflectance higher than 95%, as shown in Fig. 4d. The contribution of BNNS in ameliorating the reflectance can be best understood from its higher refractive index ($n_{BNNS} \approx 2.1$) than both WPU ($n_{WPU} \approx 1.4$) and air ($n_{air} \approx 1$)[67,68], which led to the multiscale mechanisms as discussed in the following. Macroscopically, the effective refractive index of solid pore walls was raised by adding BNNS, amplifying the difference in refractive index between solid and air so as to enhance the light scattering at their interfaces (Fig. 4e). At the microscopic scale, the contrasting refractive indices of BNNS and WPU enabled additional scattering mechanisms at the BNNS/WPU interfaces for further improved reflection (Fig. 4f). Moreover, the highly aligned 2D BNNS in the pore walls were expected to promote more backward than forward scattering of incident light, reducing the number of scattering events required to reverse the light path for improved scattering efficiencies[69]. Nanoscopically, unlike neat polymers with plenty of functional groups to absorb IR energy (Fig. 4g), BNNS contained few functional groups (Supplementary Fig. 25), leading to an improved reflectance in the NIR region (Supplementary Fig. 13b).

To summarize, our ACA achieved an excellent reflectance of 97% across the whole solar spectrum thanks to the multiple scattering of incident light by numerous interfaces created by the abundant pores and the addition of BNNS with a largely different refractive index from those of WPU and air. The highly porous structure provided large solid/air interfacial areas for multiple scattering (Fig. 4e), while the presence of 2D BNNS added to additional scattering mechanisms at the BNNS/WPU interfaces with improved backward scattering (Fig. 4f). The excellent dual functionalities of an ultralow $k$ of 16.9 mW m$^{-1}$ K$^{-1}$ and a high solar reflectance of 97% allowed the ACA to minimize both parasitic and solar heat gains by the interior environments when used

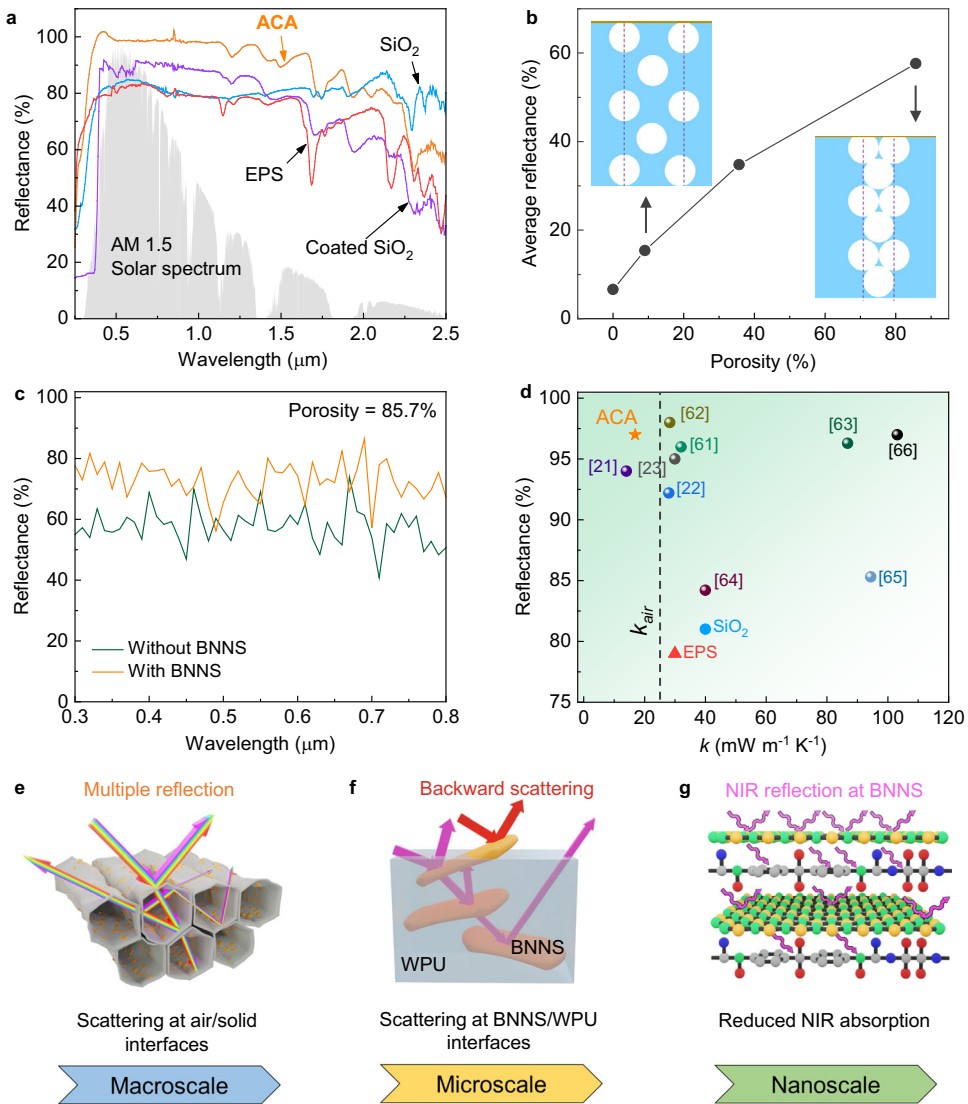

**Fig. 4 | Optical properties of ACA. a** Solar reflection spectra of commercial EPS foam, SiO₂ aerogel with and without a high-reflective coating, and ACA. The measured solar reflectances of commercial EPS foam, SiO₂ aerogel with and without a high-reflective coating, and ACA are 79%, 85%, 81% and 97%, respectively. **b** Effect of porosity on average reflectance of pristine WPU aerogels estimated from the FDTD simulations. **c** Simulated reflection spectra of highly porous WPU aerogels with and without BNNS. **d** Comparison of the solar reflectance and

thermal conductivity of ACA with those of strate-of-the-art thermally insulating materials reported in the literature[21–23,61–66] and commercial products (i.e., SiO₂ and EPS foam). Schematics of the multiscale mechanisms of extensive light scattering by (**e**) solid/air interfaces due to a highly porous structure, (**f**) BNNS/WPU interfaces because of the contrasting refractive indices, and (**g**) reduced NIR absorption because of BNNS. The atoms in orange, green, blue, red and grey colours represent boron, nitrogen, oxygen, hydrogen and carbon, respectively.

as building envelopes, offering a promising alternative to existing solutions for more energy-efficient buildings with reduced energy consumption for space cooling.

**Large-scale ACA panels for practical applications**

The scalable processing techniques involved in fabricating ACA and cost-effective raw materials are promising for the large-scale fabrication of thermal insulation panels for building envelope applications (see Supplementary Note 4, Supplementary Information for a detailed scalability analysis). To demonstrate the practical thermal insulation application of the ACA, large-size ACA panels were fabricated using the additive freeze-casting technique (Fig. 1). As shown in Supplementary Fig. 14, their lateral dimensions were 20 cm × 20 cm with a thickness of 0.7 cm. To confirm the uniformity of properties at different positions across the whole panel, the ACA panel was cut into three pieces in the alignment direction and their $k_{trans}$ values and reflectances were measured (Supplementary Fig. 15). Both properties were highly consistent across three

different regions with low standard deviations of 1.1 mW m⁻¹ K⁻¹ and 1.2% for $k_{trans}$ and reflectance, respectively (Fig. 5a), attesting to the uniform microstructure of the ACA panel produced by the additive freeze-casting (Fig. 1b). Although the properties of the large ACA panel were marginally inferior to those of the smaller samples, the $k_{trans}$ held the critical values below that of the air (24 mW m⁻¹ K⁻¹) and the reflectance remained all above 90%, demonstrating the high overall quality of the ACA panel. It should be noted that the lower $k_{trans}$ values than that of air were attributed to the consistent pore alignment of ACA thanks to the uniform ice crystal growths facilitated by additive freeze-casting (Supplementary Fig. 16a). By contrast, conventional one-step freeze-casting led to random pore alignments when the freezing front advanced away from the fixed cold source (Supplementary Fig. 16b), resulting in a high $k_{trans}$ of 28.5 ± 2.8 mW m⁻¹ K⁻¹ exceeding that of the air (Supplementary Fig. 16c). Moreover, the better alignment achieved by additive freeze-casting also offered a higher $k_{align}$ value of ACA than that made by conventional one-step freeze-casting, giving rise to a

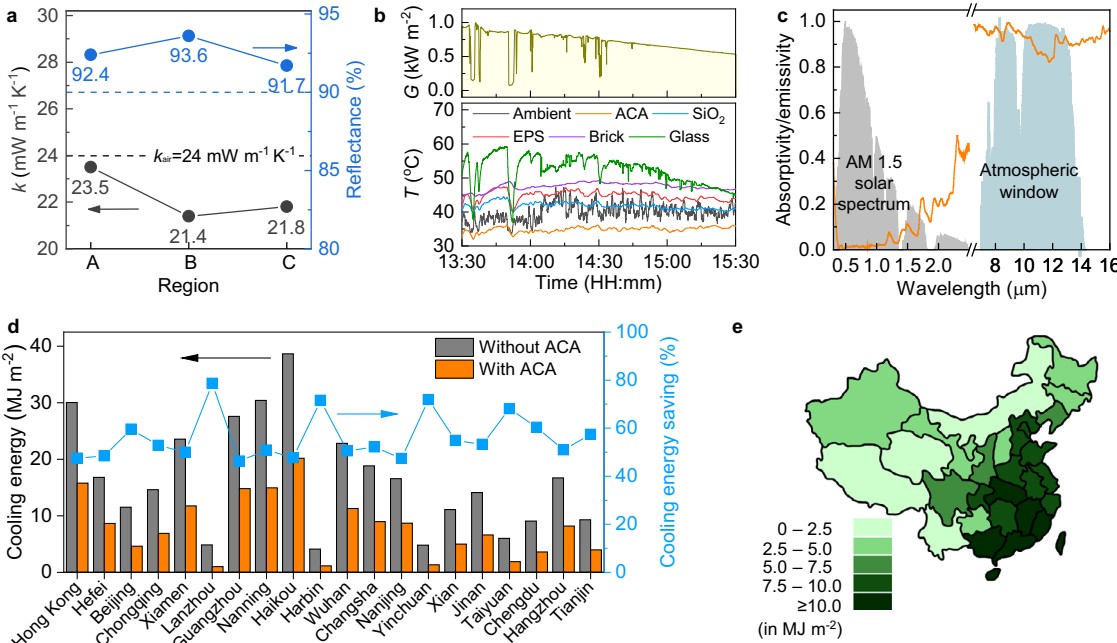

**Fig. 5 | Cooling performance of the large-scale ACA panel. a** Thermal conductivity and solar weighted reflectance of the ACA panel at different regions, showing a highly consistent performance. **b** Solar irradiance (*G*) and internal temperature (*T*) changes in different samples in the outdoor tests. **c** Selective absorptivity/emissivity spectra of the ACA in the solar and LWIR wavelengths. **d** Year-round cooling energy consumption of a building model with and without an ACA envelope and the potential cooling energy savings by using ACA in 20 different cities. **e** Year-round energy savings in different climate regions in China.

higher anisotropic factor (*R*) of the former for a superior thermal insulation performance.

The thermal insulation performance of the large ACA panel under practical conditions was evaluated by exposing the panel under direct sunlight. The outdoor tests were carried out in Hong Kong on 23rd June 2022, whose weather was hot and humid. Five samples, including our ACA panel, a brick, a transparent glass, and two commercial insulation products – EPS foam and $SiO_2$ aerogel – were placed side-by-side in five identical custom-made set-ups (Supplementary Fig. 17a). Each set-up was made of EPS foam with an open window at the top which was covered by the testing sample to form an enclosure (Supplementary Fig. 17b), and the temperature within the enclosure was monitored using a thermocouple when the sample was exposed under direct sunlight. The interior temperature under the glass cover fluctuated following the trend of solar irradiance likely due to its high thermal conductivity (0.8 W m⁻¹ K⁻¹) and transparency, allowing direct transmission of sunlight into the interior[70]. By contrast, the other opaque samples showed relatively stable internal temperatures which were much lower than that of transparent glass. It should be noted that the interior temperature under the glass panel was directly affected by the transmitted sunlight. Therefore, we kept the comparison of internal temperature to opaque materials. Overall, our ACA panel maintained the lowest interior temperature throughout the whole period among all opaque materials studied, as shown in Fig. 5b. Between 14:00 and 15:00, when the solar irradiance was 640–900 W m⁻² (Fig. 5b), our ACA panel maintained an average temperature of 35.0 °C, which was 7, 11, and 13 °C lower than the $SiO_2$ aerogel, EPS foam, and brick counterparts, respectively. A lower temperature signifies that less energy is required to maintain a cool interior environment when the ACA is used rather than the other conventional or state-of-the-art envelopes, which means a significant reduction in electricity consumption for space cooling in energy-efficient buildings. Moreover, the interior temperature under the ACA cover was even 6.1 °C lower than that of ambient on average, signifying possible passive cooling without consuming any energy when using the ACA panel as building envelope. The daytime passive cooling was enabled by a high

emissivity of 88% in the 8–13 μm atmospheric window of ACA in addition to its high reflectance, i.e., low emissivity, in the solar wavelength, as shown in Fig. 5c. This selective emissivity led to a strong long-wave IR (LWIR) emission to the cold exterior through the atmospheric window while rejecting the solar heat gain to realize passive radiative cooling[71–75]. The energy-saving potential of ACA in different climate regions was demonstrated by comparing the year-round cooling energy consumption of a building model with and without an ACA envelope using EnergyPlus (version 9.6.0) (see calculation details in Supplementary Note 6, Supplementary Information). As shown in Fig. 5d, the cooling energy consumptions for all 20 cities across China were significantly reduced when ACA was used, achieving a 56.0% cooling energy saving on average compared to the baseline without ACA. The year-round savings of cooling energy for different climate regions in China are highlighted in Fig. 5e. The application of ACA in tropical and subtropical climates had greater potential for cooling energy savings with more than 10 MJ m⁻² annual saving in southern China than in cooler temperate regions like northern China.

In real-world thermal insulation applications, the weather resistance is an important criterion for an ACA. The water contact angle of ACA was ~110° (Supplementary Fig. 20a), indicating a hydrophobic surface comparable to that of a polymer coating for radiative cooling of building envelopes[76]. The hydrophobic surface contributed to the repellent of water droplets for a good water resistance (Supplementary Fig. 20b, c and Supplementary Movie 3). The intrinsic hydrophobicity of ACA is complemented by its compatibility with a transparent waterproofing PU coating, which further increased the water contact angle to ~120° and avoided water absorption into the porous structure (Supplementary Fig. 21 and Supplementary Movie 4). The lateral surfaces of ACA were highly porous and thus prone to water uptake when exposed. The waterproof PU coating was also applied on the lateral surfaces to enhance their hydrophobicity and avoid the water transfer through the longitudinal pore channels (Supplementary Fig. 21e). Importantly, the coating showed almost no adverse effects on the ultralow thermal conductivity (Supplementary Table 2) and high solar reflectance (Supplementary Fig. 22a) of ACA thanks to its thin and

transparent nature, resulting in an internal temperature variation much the same as compared to that without coating in the outdoor test (Supplementary Fig. 22b). Moreover, the hydrophobicity also allowed easy removal of dust such as sand granules accumulated on the surface of ACA by washing with water without deteriorating its insulation performance (Supplementary Fig. 23), signifying the durability of ACA panels for long-term usage under different weather conditions[77]. The foregoing findings on the large-scale ACA panel demonstrated the effectiveness of the additive freeze-casting technique in producing solar reflective, thermally insulating aerogels for energy-efficient cooling.

## Discussion

A thermally insulating, solar-reflective ACA panel containing pore channels aligned in the plane direction with engineered pore walls was developed using BNNS/WPU composite solution by an additive freeze-casting technique. The additive freeze-casting enabled the cumulative block-by-block freezing of BNNS/WPU colloidal solution while maintaining a short freezing distance in each block, resulting in the fabrication of decimeter-scale aerogel panels having in-plane pore channels with consistent alignments and pore sizes. The incorporation of 2D BNNS offered two distinct characteristics to the additive freeze-cast ACA compared to previous thermal management solutions. First, although BNNS has been extensively used for cooling applications, most efforts were focused on utilizing the high in-plane $k$ of BNNS to dissipate the heat, e.g., for personal[78] and electronics cooling[79], while the anisotropic $k$ of BNNS was not fully exploited. Here, the consistently aligned BNNS in the pore walls by additive freeze-casting translated the anisotropic $k$ of BNNS into the ACA, leading to an ultralow $k$ in the thickness direction for effective thermal insulation. Second, the optical properties of BNNS have been mostly overlooked previously for thermal management applications. By contrast, our ACA leveraged the high refractive index of BNNS to reduce the solar heat gain, delivering an excellent solar reflectance together with an ultralow $k$. The ACA containing 10 wt% BNNS exhibited an ultralow $k$ of 16.9 mW m$^{-1}$ K$^{-1}$ in the thickness direction and a high anisotropic factor of 5.8. The porous structure provided large solid/air interfacial areas for multiple scattering at a macro-scale, while the presence of 2D BNNS having a refractive index largely different from the WPU matrix counterpart added to additional scattering mechanisms at the BNNS/WPU interfaces with improved backward scattering at a microscale. Nanoscopically, the largely different IR energy absorption capabilities of BNNS fillers and WPU matrix led to an enhanced reflectance in the NIR region. These multiscale scattering mechanisms prevailing in the ACA under sunlight contributed to a high solar reflectance of 97%. The solar reflection of ACA was among the highest radiative cooling materials while the thermal conductivity was among the lowest thermal insulation materials (Supplementary Table 3). The excellent dual functionalities enabled the ACA to lessen both parasitic and solar heat gains when used as cooling panel under direct sunlight, potentially superior to a combination of thermal insulation aerogel and radiative cooling coating to significantly reduce the energy consumption for cooling applications. The practical application was further demonstrated using a large ACA panel of 20 cm × 20 cm in lateral dimensions made by the additive freeze-casting. The ACA panel presented an ultralow $k$ (≤24 mW m$^{-1}$ K$^{-1}$) and a high solar reflectance (≥90%) consistently throughout the whole panel. Compared to commercial SiO$_2$ aerogel, the ACA provided an up to 7 °C lower interior temperature when used as the cooling panel under direct sunlight. The excellent performance of the large-scale ACA panel revealed the effectiveness of the additive freeze-casting technique in producing solar reflective, thermally insulating aerogels, offering a new paradigm for the bottom-up fabrication of scalable anisotropic aerogels from nanoscale constituents towards real-world applications.

## Methods

### Unidirectional freeze casting

The BNNS was exfoliated from bulk $h$-BN powders (3 M, Germany) by the liquid phase exfoliation (see Supplementary Method, Supplementary Information for details). The BNNS solution was sonicated in an ultrasonic bath for 30 min at 20 °C to uniformly disperse BNNS before mixing with a WPU solution (NeoRez R610 supplied by DSM NeoResin). The mixture was stirred mechanically at room temperature for 12 h before the freeze-casting process. The solution was poured into a polymer mold placed on top of a metal cold source, which was in direct contact with liquid nitrogen for unidirectional freeze-casting. The freeze-cast samples were dried in a freeze dryer (SCIENTZ-10N) at a pressure of less than 5 Pa and a temperature of −56 °C for at least 48 h to obtain aerogels.

### Additive freeze casting

For fabrication of large-scale samples, the moving cold source filled with liquid nitrogen was initially put in a position at a distance of 15 mm or less from the closed end of the Teflon mold. Once the surface temperature of the moving cold source was steady at ∼ −170 °C, the BNNS/WPU colloidal solution was dispensed into the gap between the mold end and the side of moving cold source, allowing the formation of the first block by directional freezing. After the solution was completely frozen, the moving cold source was moved in the opposite direction of freezing by the same distance. The same volume of BNNS/WPU solution was filled into the gap between the first frozen block and the side of moving cold source, and the above procedure was repeated until a desired dimension of aerogel panel was attained. To ensure a constant freezing temperature, the moving cold source was refilled with liquid nitrogen at a rate of 1 L min$^{-1}$. The freeze-cast BNNS/WPU panel was freeze-dried under the same condition as above to obtain large-scale aerogel panels.

### Characterization

The morphologies of the BNNS and aerogel samples were characterized using a SEM (Hitachi TM3030) and TEM (JEOL JEM 2010). The thickness and size of BNNS were measured on an atomic force microscope (AFM, NanoScope IIIa/Dimension 3100), as shown in Supplementary Fig. 24. The elemental compositions and chemical structure of BNNS were examined using the X-ray photoelectron spectroscopy (XPS, Axis Ultra DLD), as shown in Supplementary Fig. 25. The functional groups on BNNS, WPU aerogels, and BNNS/WPU composite aerogels were studied using the FTIR (Bruker Vertex 70 Hyperion 1000). The apparent density ($\rho$) of aerogels was calculated by weighing the samples and measuring their volumes. The porosity ($P$) of aerogels was determined by $P = \left(1 - \frac{\rho}{\rho_0}\right) \times 100\%$, where $\rho_0$ is the solid density estimated from the weighted-average of densities of WPU (1.02 g cm$^{-3}$) and BNNS (2.25 g cm$^{-3}$) for a given BNNS loading. The $k$ of samples was measured using a hot disk thermal constants analyzer (Hot Disk TPS 2500 S). The solar reflectance of samples was determined using a UV-visible NIR spectrometer (Perkin Elmer Lambda 950) in a wavelength range of 0.25–2.5 μm. The solar weighted reflectance was calculated according to the specification, ASTM G173-03. The temperature distribution was recorded using an infrared camera (Fluke Ti25).

### Outdoor tests

The custom-made set-ups for outdoor tests were constructed of EPS foam with an open window at the top covered by the testing sample to form an enclosure. All other faces of the set-up were covered using aluminum foils to avoid the absorption of solar irradiation. During the outdoor tests, the exterior surfaces of the test samples were directly exposed to the sunlight, simulating the scenario when they were used as the building envelope. A polyethylene film was placed beneath the test samples to minimize the convection between the interior and

ambient environment (Supplementary Fig. 17)[80,81]. The solar irradiance and temperature were monitored using a solar meter (TES-1333R Solar Power Meter) and a thermocouple (CENTER-309 Portable Digital Thermometer), respectively. The ambient temperature was recorded using the thermocouple covered by aluminum foil with an open structure to avoid heating from sunlight while allowing sufficient convection. The thermocouples were calibrated by CENTER Technology using the standard traceable to National Institute of Standards and Technology. The measurement errors for the outdoor tests were estimated by measuring the interior temperatures of three different ACA samples simultaneously under the same condition (Supplementary Fig. 18). They exhibited quite consistent variations in internal temperatures with an average standard deviation of $\pm 0.3\,°C$ and a maximum standard deviation not exceeding $\pm 1.0\,°C$, indicating acceptable reliability of outdoor test results.

## Numerical simulations

The FDTD simulations were performed in a wavelength range of $0.3-0.8\,\mu m$ because the energy from this region contributed over 55% to the total solar irradiance. The simulation model was built by creating longitudinal air pore channels along the Y axis, as shown in Supplementary Fig. 12a, and parallel to each other with hexagonal arrangement in the WPU film. The thickness of all models was fixed at 110 μm. The effective refractive index and extinction coefficient of the pore walls consisting of WPU and BNNS[67,68] were calculated based on the volume fraction of BNNS in the WPU aerogels, assuming perfect alignments and even distribution of BNNS in the pore alignment direction.

## Data availability

Source data are provided with this paper.

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

## Acknowledgements

This project was financially supported by the Research Grants Council (GRF Projects: 16205517 received by J.-K.K.; 16200720, received by X.S.) and the Innovation and Technology Commission (ITS/012/19, received by X.S.) of Hong Kong SAR, and start-up fund for new recruits of PolyU (P0038855, received by X.S.; P0038858, received by X.S.). Technical assistance from the Materials Characterization and Preparation Facilities (MCPF) and the Advanced Engineering Material Facility (AEMF) of HKUST are appreciated.

## Author contributions

K.-Y.C.: Conceptualization, Methodology, Data curation, Formal analysis, Writing—original draft. X.S.: Conceptualization, Formal analysis, Supervision, Funding acquisition, Writing—review & editing. J.Y.: Conceptualization, Methodology, Data curation, Writing—original draft. K.-T.L.: Methodology, Data curation, Formal analysis, Software, Writing—review & editing. H.V.: Formal analysis, Investigation. E.K.: Formal analysis, Investigation. H.Z.: Software, Formal analysis. J.-H.L.: Formal analysis, Investigation. J.Y.: Formal analysis, Resources. J.Y.: Formal analysis, Resources. J.K.K.: Supervision, Resources, Formal analysis, Funding acquisition, Writing—review & editing.

## Competing interests

K.-Y.C., X.S., J.L.Y., and J.-K.K. are inventors of a patent related to this work (application No.: 202111108494.1, filed on 22 September 2021 to China National Intellectual Property Administration). All other authors declare no competing interests.
