## [Peer Review File · Nature Communications]

Scalable anisotropic cooling aerogels by additive freeze-castingREVIEWER COMMENTS

Reviewer #1 (Remarks to the Author):

The authors presented an interesting directional freeze casting approach for fabrication of large-scale polyurethane-based aerogels composed of highly ordered micropores with uniform dimensions. Highly ordered microstructure provided an anisotropy in the thermal conductivity of the fabricated aerogels. Authors further improved the solar reflectance performance of the aerogels by using BNNS additives and achieved a thermal conductivity of 16.9 mW m⁻¹ K⁻¹ with a solar reflectance of 97%. I recommend the manuscript for the publication after addressing the following minor issues.

1. The authors mention that they achieved consistent pore sizes of 20 – 30 μm along the entire dimension in the freezing direction. They don't provide what kind of measurement they performed. Although Supplementary Fig. 1 and 2 compares the alignment of the pores obtained after the novel and conventional freeze casting approaches, these images don't provide any information regarding the consistent size of the pores. Can the authors provide SEM images taken in the plane, whose normal is parallel to the freezing direction (similar to Fig 2 a – xy plane, from different locations of your ACA)?
2. The outdoor tests the authors performed provides a great insight for the real-world thermal insulation applications, especially when the ACA is exposed to ambient conditions. Authors also should discuss if the ACA will protect its structural integrity with changing weather conditions (e.g. will the waterborne polyurethane based aerogel be stable when exposed to rain?).
3. Authors should mention the source of the BNNS solution.

Reviewer #2 (Remarks to the Author):

The manuscript with the title "Scalable Anisotropic Cooling Aerogels by Additive Freeze-Casting" and Manuscript ID: NCOMMS-22-10974 by Chan et al. The work is appropriate for the journal, there are some observations that must be addressed before the manuscript can be accepted.

- The abstract should clearly indicate the relevance of the work for international research.
- The authors should summarize the central core of knowledge that is the focus of the paper and better discuss the importance.
- The last part of the introduction should conclude the limitations of the previous studies and provide the main objectives and novelties of this study. You need to clearly address the knowledge gap and provide some meaningful phrases that your study can advance the knowledge and can fill in a knowledge gap that has not been considered yet.
- In the literature review part, you should perform a potent literature review and scrutinize the most relevant and recent published papers in high-quality journal articles. The literature review is one of the main parts of a scientific paper to show your novelty, and alert the readers that you are aware of the performed research studies.
- You need a flowchart describing the whole investigation procedure to help the readers perceive the main points. In the flowchart, describe the methods used in this study step by step and also link the methods to the corresponding results.
- Research methods should be elaborated and justified.
- Describe the methods chronologically. This is very important to help the readers to replicate your results. Please cite previous research studies where necessary.
- The figures have not been appropriately explained as well. The readers cannot perceive the main points. Please describe the critical points and trends in the figures.
- The authors should discuss the potential cause of results and not only describe that it happens. In addition, the results should be discussed more deeply in respect to other studies.
- The authors must work harder in the explanation of the results since in work, they found very interesting data that must be discussed in greater depth.
- Grammar and syntax must be improved.

Reviewer #3 (Remarks to the Author):

This paper presented an additive freeze-casting technique to fabricate thermally insulating, solar-reflective anisotropic cooling aerogel (ACA) panel containing pore channels aligned in the plane direction using boron nitride nanosheets (BNNS) and waterborne polyurethane (WPU) composite. The author demonstrated that the produced ACA has unique anisotropic thermos-optical properties, delivering simultaneously anisotropic thermal insulation and an excellent solar reflectance. The paper is very well-written and organized, it's very easy for readers to follow. The results and discussion are sufficient to support the objective of the work. The idea of additive freeze casting is technically sound, which can potentially address the scalability issue for anisotropic aerogels towards practical energy-efficient cooling applications. However, the reviewer has some concerns about the novelty and the scalability:

(1) The freeze-drying approach has been well-established in the past decades and used in various applications for fabricating porous structures. Multi-step freezing methods have also been used in the fields, even though the detailed procedures and applications are different from the proposed approach. Since the additive freeze casting was regarded as the main novelty in this paper, it would be beneficial to compare the insulation performance between the additive freeze-drying approach and the traditional one-step freeze-drying approach.

(2) The BNNS has been extensively used for thermal insulation applications (for example, personal cooling). The paper would be strengthened if the authors could highlight the uniqueness of the BNNS in the additive freeze-dried structures. Considering the building envelope is on a large scale (for example, much larger than personal cooling), it is unclear whether incorporating BNNS is a scalable approach for practical applications.

Reviewer #4 (Remarks to the Author):

This work prepared an anisotropic cooling aerogel (ACA) panel with ultralow thermal conductivity and high solar reflectance aiming to save energy consumption required for space cooling in buildings. My major comments are as follows:

1. Lines 113-117, Page 6: in addition to the four features the authors pointed out, the mechanical strength and cost of the ACA should also be considered during design and preparation. The corresponding characterization and calculation should be presented in the manuscript.

2. The weather resistance of the ACA panel is not clear. Will the solar reflectivity drop significantly with accumulative dust on the surface? How about the hydrophobicity of the material? With a high porosity, I suppose that the ACA panel would uptake rainwater which leads to a considerable increment in thermal conductivity and a remarkable decrement in solar reflectivity.

3. Fig. 5b: In addition to the four materials, the typical building wall materials (e.g., brick) should be involved as the base case.

4. Fig. 5e: Why the fluctuation of the glass temperature was much more significant than that of the other three panels? Is this because the thermocouple probe attached to the glass is exposed to the sunlight directly? Typically, the thermocouple probe should be covered by aluminum foil to prevent exposure to the sunlight.

5. Lines 415-416, Page 17: Why use a PE film atop the ACA panel? The panel itself should be the exterior surface of the building envelope in the real-world application (just as the authors illustrated in Fig. 1a).

6. Besides, calibration of the thermocouples is needed. Error analysis of the outdoor testing results should also be included.

7. A simple comparison of stagnation temperatures among the four selected materials is inadequate to demonstrate the energy-saving potential of the ACA material. Building energy simulation is a necessity to quantitatively predict the year-round potential impact of the ACA on building energy efficiency in different climate regions.

8. Overall, the proposed scheme, namely, developing an aerogel-based building envelope material with ultralow thermal conductivity and high solar reflectance for building energy-saving, is not novel enough. The performance of the large-scale ACA material is also not sufficiently good.

Besides, one can easily achieve the goal proposed in this work by combining an aerogel with even lower thermal conductivity and a top layer of radiative cooling coating with higher solar reflectance and much better weather resistance.

Authors' response to Reviewers' Comments:

Reviewer #1

The authors presented an interesting directional freeze casting approach for fabrication of large-scale polyurethane-based aerogels composed of highly ordered micropores with uniform dimensions. Highly ordered microstructure provided an anisotropy in the thermal conductivity of the fabricated aerogels. Authors further improved the solar reflectance performance of the aerogels by using BNNS additives and achieved a thermal conductivity of $16.9 \text{ mW m}^{-1} \text{ K}^{-1}$ with a solar reflectance of 97%. I recommend the manuscript for the publication after addressing the following minor issues.

1. The authors mention that they achieved consistent pore sizes of $20 - 30 \mu\text{m}$ along the entire dimension in the freezing direction. They don't provide what kind of measurement they performed. Although Supplementary Fig. 1 and 2 compares the alignment of the pores obtained after the novel and conventional freeze casting approaches, these images don't provide any information regarding the consistent size of the pores. Can the authors provide SEM images taken in the plane, whose normal is parallel to the freezing direction (similar to Fig 2 a – xy plane, from different locations of your ACA)?

Response:

As suggested, SEM images showing the cross-sections at different locations along the freezing direction and detailed measurements of pore size distribution are now included in the revision:

Page 7:

“The SEM images taken from different frozen blocks show in-plane aligned pore channels (Fig. 1c and Supplementary Fig. 1), in contrast to the randomly aligned pores in the aerogel obtained by conventional unidirectional freeze-casting (Supplementary Fig. 2). The pore sizes were measured from the cross-sectional SEM images taken at different locations along the freezing direction, indicating highly consistent pore sizes of $\sim 30 \mu\text{m}$ along the entire dimension (Supplementary Fig. 3).”

Supplementary Fig. 3:

“

Supplementary Fig. 3. (a-d) Cross-sectional SEM images taken at different locations along the freezing direction of the WPU aerogel made by additive freeze-casting. (e-f) The corresponding pore size distributions measured from a-d, respectively, using Nano measurer software. The fitting with Gaussian distribution indicates a consistent mean pore size of $\sim 30 \mu\text{m}$ at different distances from the cold source.”

2. The outdoor tests the authors performed provides a great insight for the real-world thermal insulation applications, especially when the ACA is exposed to ambient conditions. Authors also should discuss if the ACA will protect its structural integrity with changing weather conditions (e.g. will the waterborne polyurethane based aerogel be stable when exposed to rain?).

Response:

To understand whether the raindrops would affect the structural integrity of ACA aerogel, the water contact angle of ACA was first measured to understand its hydrophobicity. As shown in Supplementary Fig. 20a, the water contact angle of ACA was $\sim 110^\circ$, indicating a hydrophobic surface. This contact angle value is comparable to that of a porous polymer coating for radiative cooling of building envelopes (*Science* 2018, **362**, 315-319). The hydrophobic surface was beneficial to the repellent of water droplets under raining conditions, as demonstrated in Supplementary Fig. 20b and Supplementary Video 2. The water droplet easily rolled off from the ACA surface, which remained dry with no noticeable changes after being washed by dyed water droplets continuously for 5 min (Supplementary Fig. 20c), suggesting a good water resistance of ACA. We note that the ACA is also compatible with commercial transparent waterproofing PU coatings without affecting its excellent thermo-optical properties, as shown in Supplementary Figs. 21-22 and Supplementary Table 2.

The above discussion is included in the revision:

Page 16:

“In real-world thermal insulation applications, the weather resistance is an important criterion for an ACA. The water contact angle of ACA was $\sim 110^\circ$ (Supplementary Fig. 20a), indicating

a hydrophobic surface comparable to that of a polymer coating for radiative cooling of building envelopes.⁷⁶ The hydrophobic surface contributed to the repellent of water droplets for a good water resistance (Supplementary Fig. 20b-c and Supplementary Video 2). The intrinsic hydrophobicity of ACA is complemented by its compatibility with a transparent waterproofing PU coating, which further increased the water contact angle to $\sim 120^\circ$ and avoided water absorption into the porous structure (Supplementary Fig. 21 and Supplementary Video 3). Importantly, the coating showed almost no adverse effects on the ultralow thermal conductivity (Supplementary Table 2) and high solar reflectance (Supplementary Fig. 22a) of ACA thanks to its thin and transparent nature, resulting in an internal temperature variation much the same as compared to that without coating in the outdoor test (Supplementary Fig. 22b).”

References:

“76. Mandal J, et al. Hierarchically porous polymer coatings for highly efficient passive daytime radiative cooling. *Science* **362**, 315-319 (2018).”

Supplementary Information:

“S7. Weather resistance of ACA

Supplementary Fig. 20. Water resistance of ACA. (a) Water contact angle of ACA. (b) Photograph showing a snapshot from the water resistance test of ACA. The whole test is shown in Supplementary Video 2. (c) Photograph showing the surface of ACA after being washed with dyed water for 5 min. The surface of ACA remained intact with no noticeable changes, suggesting a good water resistance of ACA thanks to its hydrophobic surface.

Supplementary Fig. 21. Water resistance of PU-coated ACA. (a) Cross-sectional SEM image of the PU-coated ACA (scale bar: 50 μm). The thickness of PU coating is $\sim 80 \mu\text{m}$. (b) Surface SEM images of PU-coated ACA compared to that without coating (scale bars: 20 μm). (c) Water contact angle of PU-coated ACA. (d) Photographs showing the changes of water droplets on the surfaces of ACA with and without PU coating with time.

The ACA is compatible with commercial waterproofing coatings without losing the excellent thermal and optical properties. A transparent waterproofing coating (JY-S66, Shanghai Hanlong Company) for building envelopes was applied on the ACA surface via spray-coating. The PU coating had a thickness of $\sim 80 \mu\text{m}$ (Supplementary Fig. 21a), blocking the surface pores of ACA (Supplementary Fig. 21b) to avoid water uptake through these pores. As shown in Supplementary Fig. 21c, the PU coating enhanced the water contact angle of ACA to 120° , making it highly water repellent (Supplementary Video 3). Moreover, the water droplets applied on the coated surface remained unchanged after 60 minutes, demonstrating an excellent waterproof characteristic (Supplementary Fig. 21d). By contrast, the water droplets were gradually absorbed into the ACA without coating through its surface pores (Supplementary Fig. 21d).

Supplementary Table 2. Physical and thermal properties of ACA and PU-coated ACA.

Sample	Density (mg cm^{-3})	Porosity (%)	Thermal conductivity ($\text{mW m}^{-1} \text{K}^{-1}$)	
			Transverse	Alignment
ACA	20.2 ± 1.3	98.2 ± 0.1	17.0 ± 0.6	98.4 ± 7.2
PU-coated ACA	22.0 ± 1.0	98.0 ± 0.1	18.6 ± 0.7	101.3 ± 3.9

Supplementary Fig. 22. (a) Solar reflection spectra of ACA and PU-coated ACA. (b) Solar irradiance, G , and internal temperature changes in ACA, PU-coated ACA, and PU-coated ACA after wetting during the outdoor tests. For the last sample, the surface was continuously wetted by water droplets for 5 min as demonstrated in Supplementary Video 3 prior to the outdoor test.

The waterproofing PU coating only marginally increased the thermal conductivity (Supplementary Table 2) while it had no adverse effect on the high solar reflectance (Supplementary Fig. 22a) of ACA thanks to its thin and transparent nature. Outdoor tests were also performed to compare the insulation performance of ACA and PU-coated ACA under practical conditions. The PU-coated ACA showed an internal temperature variation much the same as that without coating in the outdoor test even after wetting prior to the test (Supplementary Fig. 22b), showing a negligible effect of PU coating on the practical insulation performance.”

3. Authors should mention the source of the BNNS solution.

Response:

As suggested, the source of BNNS solution has been included in the revision:

Page 18:

“The BNNS was exfoliated from bulk *h*-BN powders (3M, Germany) by the liquid phase exfoliation (see Section S9, Supplementary Information for details).”

Reviewer #3

This paper presented an additive freeze-casting technique to fabricate thermally insulating, solar-reflective anisotropic cooling aerogel (ACA) panel containing pore channels aligned in the plane direction using boron nitride nanosheets (BNNS) and waterborne polyurethane (WPU) composite. The author demonstrated that the produced ACA has unique anisotropic thermo-optical properties, delivering simultaneously anisotropic thermal insulation and an excellent solar reflectance. The paper is very well-written and organized, it’s very easy for readers to follow. The results and discussion are sufficient to support the objective of the work. The idea of additive freeze casting is technically sound, which can potentially address the scalability issue for anisotropic aerogels towards practical energy-efficient cooling applications. However, the reviewer has some concerns about the novelty and the scalability:

(1) The freeze-drying approach has been well-established in the past decades and used in

various applications for fabricating porous structures. Multi-step freezing methods have also been used in the fields, even though the detailed procedures and applications are different from the proposed approach. Since the additive freeze casting was regarded as the main novelty in this paper, it would be beneficial to compare the insulation performance between the additive freeze-drying approach and the traditional one-step freeze-drying approach.

Response:

The authors appreciate the critical point raised by the reviewer. Although multistep freeze-casting has been reported previously (e.g., *J Euro Ceram Soc* 2020, **40**, 1398-1406; *Rev Sci Instrum* 2020, **91**, 033904; *Ceram Int* 2013, **39**, 9703-9707; *ACS Appl Mater Interfaces* 2015, **7**, 14439-14445), the currently established methods are rather difficult to scale up for producing well aligned decimeter-long pores, limiting practical application of the porous structures thereby produced. In this method, the stepwise ice growth was generated from a fixed cold source while a changing cold source temperature was used to maintain the ice front temperature for consistent alignment (*J Euro Ceram Soc* 2020, **40**, 1398-1406). Nonetheless, because the lowest temperature was limited by the fixed cold source, the total length of the aligned pores remained very short, only a few centimeters at best. In sharp contrast, we adopted a moving cold source in our additive freeze-casting approach which enabled centimeter-long freezing distance with consistent pore alignment. In addition, the temperature at the moving cold source remained fairly constant, reducing the complexity for temperature control compared to the previous multistep freezing technique. Moreover, our additive freeze-casting allowed the continuous freezing of colloidal solution in a block-by-block manner, more efficient for large-scale fabrication than the conventional methods.

As suggested by the reviewer, we also compared the insulation performance of two aerogel samples made by additive freeze-casting and traditional one-step freeze-casting, as shown in Supplementary Fig. 16. Compared to the additive freeze-cast aerogel with a low k_{trans} of $21.6 \pm 1.1 \text{ mW m}^{-1} \text{ K}^{-1}$, a much higher k_{trans} of $28.5 \pm 2.8 \text{ mW m}^{-1} \text{ K}^{-1}$ exceeding that of the air was observed in the one-step freeze-cast aerogel, demonstrating the advantage of additive freeze-casting in achieving better insulation performance.

Based on the above discussion, the advantages of additive freeze-casting compared to conventional one-step and multi-step freezing methods are included in the revision:

Page 14:

“It should be noted that the lower k_{trans} values than that of air were attributed to the consistent pore alignment of ACA thanks to the uniform ice crystal growths facilitated by additive freeze-casting (Supplementary Fig. 16a). By contrast, conventional one-step freeze-casting led to random pore alignments when the freezing front advanced away from the fixed cold source (Supplementary Fig. 16b), resulting in a high k_{trans} of $28.5 \pm 2.8 \text{ mW m}^{-1} \text{ K}^{-1}$ exceeding that of the air (Supplementary Fig. 16c). Moreover, the better alignment achieved by additive freeze-casting also offered a higher k_{align} value of ACA than that made by conventional one-step freeze-casting, giving rise to a higher anisotropic factor (R) of the former for a superior thermal insulation performance.”

Supplementary Fig. 16:

“

Supplementary Fig. 16. Schematics showing the ice crystal growths in (a) additive freeze-casting and (b) conventional one-step unidirectional freeze-casting. (c) Comparison of thermal conductivities and anisotropic factors of two aerogels made by different freeze-casting methods.

It should be noted that the complete freezing of decimeter-scale aerogels using one-step unidirectional freeze-casting could be extremely difficult, if not entirely impossible, because the temperature of solidification front tended to rise substantially once far away from the cold source. Multistep freeze-casting methods have been reported previously,^{5,6,7,8} the stepwise ice growth was generated from a fixed cold source, limiting the overall length of the aligned pores to only a few centimeters.⁵ To compare the thermal insulation performance, the aerogels were produced by both additive freeze-casting and conventional one-step freeze-casting with the same total freezing distance of 4.5 cm using moving and fixed cold sources, respectively.”

Supplementary References:

5. Christiansen CD, Nielsen KK, Bjørk R. Functionally graded multi-material freeze-cast structures with continuous microchannels. *J Eur Ceram Soc* **40**, 1398-1406 (2020).
6. Christiansen CD, Nielsen KK, Bjørk R. Novel freeze-casting device with high precision thermoelectric temperature control for dynamic freezing conditions. *Rev Sci Instrum* **91**, 033904 (2020).
7. Tang Y, Zhao K, Hu L, Wu Z. Two-step freeze casting fabrication of hydroxyapatite porous scaffolds with bionic bone graded structure. *Ceram Int* **39**, 9703-9707 (2013).
8. Ouyang A, *et al.* Highly porous core-shell structured graphene-chitosan beads. *ACS Appl Mater Interfaces* **7**, 14439-14445 (2015).”

(2) The BNNS has been extensively used for thermal insulation applications (for example, personal cooling). The paper would be strengthened if the authors could highlight the uniqueness of the BNNS in the additive freeze-dried structures. Considering the building

envelope is on a large scale (for example, much larger than personal cooling), it is unclear whether incorporating BNNS is a scalable approach for practical applications.

Response:

As suggested by the reviewer, we have further highlighted the uniqueness of BNNS in the additive freeze-cast structures in the revision:

Page 16:

“The incorporation of 2D BNNS offered two distinct characteristics to the additive freeze-cast ACA compared to previous thermal management solutions. First, although BNNS has been extensively used for cooling applications, most efforts were focused on utilizing the high in-plane k of BNNS to dissipate the heat, e.g., for personal⁷⁸ and electronics cooling,⁷⁹ while the anisotropic k of BNNS was not fully exploited. Here, the consistently aligned BNNS in the pore walls by additive freeze-casting translated the anisotropic k of BNNS into the ACA, leading to an ultralow k in the thickness direction for effective thermal insulation. Second, the optical properties of BNNS have been mostly overlooked previously for thermal management applications. By contrast, our ACA leveraged the high refractive index of BNNS to reduce the solar heat gain, delivering an excellent solar reflectance together with an ultralow k .”

References:

78. Miao D, Wang X, Yu J, Ding B. A biomimetic transpiration textile for highly efficient personal drying and cooling. *Adv Funct Mater* **31**, 2008705 (2021).

79. Chen J, Huang X, Sun B, Jiang P. Highly thermally conductive yet electrically insulating polymer/boron nitride nanosheets nanocomposite films for improved thermal management capability. *ACS Nano* **13**, 337-345 (2018).”

In addition, the scalability analysis and materials cost estimation for large-scale fabrication of ACA have been included in the revision:

Page 13:

“The scalable processing techniques involved in fabricating ACA and cost-effective raw materials are promising for the large-scale fabrication of thermal insulation panels for building envelope applications (see Section S4, Supplementary Information for a detailed scalability analysis).”

Section S4, Supplementary Information:

“S4. Scalability analysis and materials cost estimation

The processing techniques involved in the synthesis of BNNS and additive freeze-casting, including liquid phase exfoliation, freeze-casting, and freeze-drying, are either industrially available techniques or easily scaled up for mass production.

First, the current work used the liquid phase exfoliation technique to attain BNNS from h -BN by ultrasonication, which is already well established for scalable production of BNNS in large quantities.³ Second, the freeze-drying technique employed in this work to obtain aerogels is an industrially available technique widely used in pharmaceutical and food industries. Industry-scale freeze dryers having meter-scale chambers can accommodate large-scale products made from additive freeze-casting. Third, the additive freeze-casting technique established in this work can be scaled up for fabricating decimeter- or even meter-scale aerogels by increasing the lateral dimensions of the mold and the total freezing distance. The demonstration of large-scale fabrication of a decimeter-scale ACA panel is presented in Supplementary Fig. 14.

The whole process involved in additive freeze-casting is simple and does not require complicated equipment/apparatus or expensive solvents. Therefore, the cost for scaling up the freeze-casting set-up is considered low.⁴ In addition, the raw materials used to fabricate ACA panels are available at cheap prices, as shown in Supplementary Table 1.

In view of the above analysis, the ACA developed in this work has a high potential for commercialization.

Supplementary Table 1. Cost estimation of raw materials for producing ACA panels.

Raw material	Price	Amount per m ²	Price per m ²
	US\$5.66 kg ⁻¹		
WPU resin	(https://www.alibaba.com/product-detail/Waterborne-Polyurethane-Resin-Waterborne-Polyurethane-Hydroxyl_1600356457625.html?spm=a2700.galleryofferlist.normal_offer.d_title.7b913a5bBJ9gSJ&s=p)	0.18 kg	US\$1.02
	US\$50 kg ⁻¹		
BNNS	(https://www.alibaba.com/product-detail/Good-price-Boron-Nitride-Nanosheets-BN_1600430750000.html?spm=a2700.galleryofferlist.normal_offer.d_image.2d0c33adD6JZTv)	0.02 kg	US\$1.00
Total:			US\$2.02

Note: The amounts per 1 m² of area are estimated based on the density of ACA (20.2 mg cm⁻³), the composition of 90wt% WPU and 10wt% BNNS, and the thickness of 1 cm.”

Supplementary References:

3. Coleman JN, et al. Two-dimensional nanosheets produced by liquid exfoliation of layered materials. *Science* **331**, 568-571 (2011).
4. Yu Z-L, et al. Bioinspired polymeric woods. *Sci Adv* **4**, eaat7223 (2018).”

Reviewer #4

This work prepared an anisotropic cooling aerogel (ACA) panel with ultralow thermal conductivity and high solar reflectance aiming to save energy consumption required for space cooling in buildings. My major comments are as follows:

1. Lines 113-117, Page 6: in addition to the four features the authors pointed out, the mechanical strength and cost of the ACA should also be considered during design and preparation. The corresponding characterization and calculation should be presented in the manuscript.

Response:

As suggested by the reviewer, we have included the mechanical properties and cost as desired features for ACA with detailed discussions in the revision.

Page 6:

“We designed ... (iv) mechanically flexible and robust to be rolled up for easy storage and transportation, and (v) cost-effective materials and processing.”

The mechanical properties are measured and discussed in detail in the below:

Page 11:

“Similar to the heat conduction, the mechanical properties of ACA were also anisotropic due to the anisotropic structure (Supplementary Fig. 9). Both the compressive modulus and strength of aerogels increased with increasing BNNS loading as a reinforcement effect of BNNS⁵⁶ with strengths of 3.71 and 7.37 kPa in the transverse and alignment directions, respectively. These values compare well with those of common aerogels, such as covalently crosslinked polymer aerogels,³¹ graphene aerogels,⁵⁷ and MXene aerogels.^{58,59}”

References:

31. Cheng Y, et al. Super-elasticity at 4 K of covalently crosslinked polyimide aerogels with negative Poisson’s ratio. *Nat Commun* **12**, 1-12 (2021).

56. Falin A, et al. Mechanical properties of atomically thin boron nitride and the role of interlayer interactions. *Nat Commun* **8**, 1-9 (2017).

57. Wan Y-J, Zhu P-L, Yu S-H, Sun R, Wong C-P, Liao W-H. Ultralight, super-elastic and volume-preserving cellulose fiber/graphene aerogel for high-performance electromagnetic interference shielding. *Carbon* **115**, 629-639 (2017).

58. Cai C, Wei Z, Huang Y, Fu Y. Wood-inspired superelastic MXene aerogels with superior photothermal conversion and durable superhydrophobicity for clean-up of super-viscous crude oil. *Chem Eng J* **421**, 127772 (2021).

59. Han M, et al. Anisotropic MXene aerogels with a mechanically tunable ratio of electromagnetic wave reflection to absorption. *Adv Opt Mater* **7**, 1900267 (2019).”

Supplementary Fig. 9:

“

Supplementary Fig. 9. Compressive (a) moduli and (b) strengths at 50% of strains of aerogels with different BNNS loadings. The uniaxial compression tests were carried out on a universal testing machine (MTS Alliance RT-5) at a crosshead speed of 2 mm min⁻¹ in accordance with ASTM – C165 – 07.”

The scalability analysis and materials cost estimation are discussed below:

Page 13:

“The scalable processing techniques involved in fabricating ACA and cost-effective raw materials are promising for the large-scale fabrication of thermal insulation panels for building envelope applications (see Section S4, Supplementary Information for a detailed scalability analysis).”

Section 4, Supplementary Information:

“S4. Scalability analysis and materials cost estimation

The processing techniques involved in the synthesis of BNNS and additive freeze-casting, including liquid phase exfoliation, freeze-casting, and freeze-drying, are either industrially available techniques or easily scaled up for mass production.

First, the current work used the liquid phase exfoliation technique to attain BNNS from h-BN by ultrasonication, which is already well established for scalable production of BNNS in large quantities.³ Second, the freeze-drying technique employed in this work to obtain aerogels is an industrially available technique widely used in pharmaceutical and food industries. Industry-scale freeze dryers having meter-scale chambers can accommodate large-scale products made from additive freeze-casting. Third, the additive freeze-casting technique established in this work can be scaled up for fabricating decimeter- or even meter-scale aerogels by increasing the lateral dimensions of the mold and the total freezing distance. The demonstration of large-scale fabrication of a decimeter-scale ACA panel is presented in Supplementary Fig. 14.

The whole process involved in additive freeze-casting is simple and does not require complicated equipment/apparatus or expensive solvents. Therefore, the cost for scaling up the freeze-casting set-up is considered low.⁴ In addition, the raw materials used to fabricate ACA panels are available at cheap prices, as shown in Supplementary Table 1.

In view of the above analysis, the ACA developed in this work has a high potential for commercialization.

Supplementary Table 1. Cost estimation of raw materials for producing ACA panels.

Raw material	Price	Amount per m ²	Price per m ²
	US\$5.66 kg ⁻¹		
WPU resin	(https://www.alibaba.com/product-detail/Waterborne-Polyurethane-Resin-Waterborne-Polyurethane-Hydroxyl_1600356457625.html?spm=a2700.galeryofferlist.normal_offer.d_title.7b913a5bBJ9gSJ&s=p)	0.18 kg	US\$1.02
BNNS	US\$50 kg ⁻¹	0.02 kg	US\$1.00

(https://www.alibaba.com/product-detail/Good-price-Boron-Nitride-Nanosheets-BN_1600430750000.html?spm=a2700.galleryof ferlist.normal_offer.d_image.2d0c33adD6JZTv)

Total: US\$2.02

Note: The amounts per 1 m² of area are estimated based on the density of ACA (20.2 mg cm⁻³), the composition of 90wt% WPU and 10wt% BNNS, and the thickness of 1 cm.”

Supplementary References:

“3. Coleman JN, et al. Two-dimensional nanosheets produced by liquid exfoliation of layered materials. *Science* **331**, 568-571 (2011).

4. Yu Z-L, et al. Bioinspired polymeric woods. *Sci Adv* **4**, eaat7223 (2018).”

2. The weather resistance of the ACA panel is not clear. Will the solar reflectivity drop significantly with accumulative dust on the surface? How about the hydrophobicity of the material? With a high porosity, I suppose that the ACA panel would uptake rainwater which leads to a considerable increment in thermal conductivity and a remarkable decrement in solar reflectivity.

Response:

As suggested by the reviewer, we have further investigated the weather resistance of the ACA panel. First, the water contact angle of ACA was measured to understand its hydrophobicity. As shown in Supplementary Fig. 20a, the water contact angle of ACA was ~110°, indicating a hydrophobic surface. This contact angle value is comparable to that of a porous polymer coating for radiative cooling of building envelopes (*Science* 2018, **362**, 315-319). The hydrophobic surface was beneficial to the repellent of water droplets under raining conditions, as demonstrated in Supplementary Fig. 20b and Supplementary Video 2. The water droplet easily rolled off from the ACA surface, which remained dry with no noticeable changes after being washed by dyed water droplets continuously for 5 min (Supplementary Fig. 20c), suggesting a good water resistance of ACA.

Second, our ACA is also compatible with commercial waterproofing coatings without losing the excellent thermal and optical properties. This trait is crucial to avoiding water uptake by ACA because water is detrimental to its insulation performance. A transparent waterproofing coating (JY-S66, Shanghai Hanlong Company) commonly used for building envelopes was applied on the ACA surface via spray-coating. The PU coating had a thickness of ~ 80 μm, blocking the surface pores of ACA to avoid water uptake through these pores (Supplementary Fig. 21a-b). As shown in Supplementary Fig. 21c, the water contact angle was improved to 120° when the coating was applied, leading to excellent water repellent (Supplementary Video 3). Moreover, the water droplets applied on the coated surface remained unchanged after 60 minutes, indicating an excellent waterproof characteristic (Supplementary Fig. 21d). By contrast, the water droplets were gradually absorbed into the ACA without coating through its surface porosity (Supplementary Fig. 21d). Importantly, the waterproofing PU coating showed negligible adverse effects on the ultralow thermal conductivity (Supplementary Table 2) and high solar reflectance (Supplementary Fig. 22a) of ACA because of its thin and transparent nature. For practical thermal insulation applications, the PU-coated ACA showed performance much the same as that without coating in the outdoor test even after wetting with water shower prior to the test (Supplementary Fig. 22b).

Third, the accumulation of dust could pose a challenge to the insulation performance of ACA as it reduces the solar reflectance performance. The common approach designed to address dust accumulation encompasses hydrophobic surfaces to allow easy removal of dust through physical cleaning (*Joule* 2020, **4**, 1350-1356). Thanks to the hydrophobic surface of PU-coated ACA, the sand granules sprinkled on the ACA surface to simulate accumulated dust were easily washed away by running water (Supplementary Fig. 23a). The undamaged surface of the PU-coated ACAs obtained after sand removal gave rise to internal temperatures quite similar to those recorded before sand removal in the outdoor tests (Supplementary Fig. 23b), signifying the durability of ACA panels for long-term usage under different weather conditions.

Based on the above, the detailed discussion on the weather resistance has been added in the revision:

Page 16:

“In real-world thermal insulation applications, the weather resistance is an important criterion for an ACA. The water contact angle of ACA was $\sim 110^\circ$ (Supplementary Fig. 20a), indicating a hydrophobic surface comparable to that of a polymer coating for radiative cooling of building envelopes.⁷⁶ The hydrophobic surface contributed to the repellent of water droplets for a good water resistance (Supplementary Fig. 20b-c and Supplementary Video 2). The intrinsic hydrophobicity of ACA is complemented by its compatibility with a transparent waterproofing PU coating, which further increased the water contact angle to $\sim 120^\circ$ and avoided water absorption into the porous structure (Supplementary Fig. 21 and Supplementary Video 3). Importantly, the coating showed almost no adverse effects on the ultralow thermal conductivity (Supplementary Table 2) and high solar reflectance (Supplementary Fig. 22a) of ACA thanks to its thin and transparent nature, resulting in an internal temperature variation much the same as compared to that without coating in the outdoor test (Supplementary Fig. 22b). Moreover, the hydrophobicity also allowed easy removal of dust such as sand granules accumulated on the surface of ACA by washing with water without deteriorating its insulation performance (Supplementary Fig. 23), signifying the durability of ACA panels for long-term usage under different weather conditions.⁷⁷”

References:

“76. Mandal J, et al. Hierarchically porous polymer coatings for highly efficient passive daytime radiative cooling. *Science* **362**, 315-319 (2018).

77. Mandal J, Yang Y, Yu N, Raman AP. Paints as a scalable and effective radiative cooling technology for buildings. *Joule* **4**, 1350-1356 (2020).”

Supplementary Information:

“S7. Weather resistance of ACA

Supplementary Fig. 20. Water resistance of ACA. (a) Water contact angle of ACA. (b) Photograph showing a snapshot from the water resistance test of ACA. The whole test is shown in Supplementary Video 2. (c) Photograph showing the surface of ACA after being washed with dyed water for 5 min. The surface of ACA remained intact with no noticeable changes, suggesting a good water resistance of ACA thanks to its hydrophobic surface.

Supplementary Fig. 21. Water resistance of PU-coated ACA. (a) Cross-sectional SEM image of the PU-coated ACA (scale bar: 50 μm). The thickness of PU coating is $\sim 80 \mu\text{m}$. (b) Surface SEM images of PU-coated ACA compared to that without coating (scale bars: 20 μm). (c) Water contact angle of PU-coated ACA. (d) Photographs showing the changes of water droplets on the surfaces of ACA with and without PU coating with time.

The ACA is compatible with commercial waterproofing coatings without losing the excellent thermal and optical properties. A transparent waterproofing coating (JY-S66, Shanghai Hanlong Company) for building envelopes was applied on the ACA surface via spray-coating. The PU coating had a thickness of $\sim 80 \mu\text{m}$ (Supplementary Fig. 21a), blocking the surface pores of ACA (Supplementary Fig. 21b) to avoid water uptake through these pores. As shown in Supplementary Fig. 21c, the PU coating enhanced the water contact angle of ACA to 120° , making it highly water repellent (Supplementary Video 3). Moreover, the water droplets applied on the coated surface remained unchanged after 60 minutes, demonstrating an excellent

waterproof characteristic (Supplementary Fig. 21d). By contrast, the water droplets were gradually absorbed into the ACA without coating through its surface pores (Supplementary Fig. 21d).

Supplementary Table 2. Physical and thermal properties of ACA and PU-coated ACA.

Sample	Density (mg cm ⁻³)	Porosity (%)	Thermal conductivity (mW m ⁻¹ K ⁻¹)	
			Transverse	Alignment
ACA	20.2 ± 1.3	98.2 ± 0.1	17.0 ± 0.6	98.4 ± 7.2
PU-coated ACA	22.0 ± 1.0	98.0 ± 0.1	18.6 ± 0.7	101.3 ± 3.9

Supplementary Fig. 22. (a) Solar reflection spectra of ACA and PU-coated ACA. (b) Solar irradiance, G , and internal temperature changes in ACA, PU-coated ACA, and PU-coated ACA after wetting during the outdoor tests. For the last sample, the surface was continuously wetted by water droplets for 5 min as demonstrated in Supplementary Video 3 prior to the outdoor test.

The waterproofing PU coating only marginally increased the thermal conductivity (Supplementary Table 2) while it had no adverse effect on the high solar reflectance (Supplementary Fig. 22a) of ACA thanks to its thin and transparent nature. Outdoor tests were also performed to compare the insulation performance of ACA and PU-coated ACA under practical conditions. The PU-coated ACA showed an internal temperature variation much the same as that without coating in the outdoor test even after wetting prior to the test (Supplementary Fig. 22b), showing a negligible effect of PU coating on the practical insulation performance.

Supplementary Fig. 23. (a) Photographs showing the surface of ACA before and after removal of sand granules by physical cleaning with water. (b) Solar irradiance, G , and internal temperature changes in PU-coated ACA before and after sand removal in the outdoor tests.

The accumulation of dust could pose a challenge to the insulation performance of ACA as it reduces the solar reflectance performance. The common approach designed to address dust accumulation encompasses hydrophobic surfaces to allow easy removal of dust through physical cleaning.¹² The sand granules sprinkled on the PU-coated ACA surface to simulate accumulated dust were easily washed away by running water owing to its hydrophobic nature (Supplementary Fig. 23a). The undamaged surface obtained after sand removal gave rise to internal temperatures quite similar to those recorded before sand removal in the outdoor tests (Supplementary Fig. 23b), signifying the high durability of ACA panels for long-term usage under different weather conditions.”

Supplementary References:

“12. Mandal J, Yang Y, Yu N, Raman AP. Paints as a scalable and effective radiative cooling technology for buildings. *Joule* **4**, 1350-1356 (2020).”

3. Fig. 5b: In addition to the four materials, the typical building wall materials (e.g., brick) should be involved as the base case.

Response:

As suggested by the reviewer, we have redone the outdoor tests after including a brick sample (Supplementary Fig. 17). The detailed comparison has been included in the revision:

Page 14:

“The thermal insulation performance of the large ACA panel under practical conditions was evaluated by exposing the panel under direct sunlight. The outdoor tests were carried out in Hong Kong on 23rd June 2022, whose weather was hot and humid. Five samples, including our ACA panel, a brick, a transparent glass, and two commercial insulation products – EPS foam and SiO₂ aerogel – were placed side-by-side in five identical custom-made set-ups (Supplementary Fig. 17a). Each set-up was made of EPS foam with an open window at the top which was covered by the testing sample to form an enclosure (Supplementary Fig. 17b), and the temperature within the enclosure was monitored using a thermocouple when the sample was exposed under direct sunlight. The interior temperature under the glass cover fluctuated following the trend of solar irradiance likely due to its high thermal conductivity ($0.8 \text{ W m}^{-1} \text{ K}^{-1}$) and transparency, allowing direct transmission of sunlight into the interior.⁷⁰ By contrast, the

other opaque samples showed relatively stable internal temperatures which were much lower than that of transparent glass. Overall, our ACA panel maintained the lowest interior temperature throughout the whole period among all materials studied, as shown in Fig. 5b. Between 14:00 and 15:00, when the solar irradiance was 640 – 900 W m⁻² (Fig. 5b), our ACA panel maintained an average temperature of 35.0 °C, which was 7, 11, 13 and 19 °C lower than the SiO₂ aerogel, EPS foam, brick and glass cover counterparts, respectively. A lower temperature signifies that less energy is required to maintain a cool interior environment when the ACA is used rather than the other conventional or state-of-the-art envelopes, which means a significant reduction in electricity consumption for space cooling in energy-efficient buildings. Moreover, the interior temperature under the ACA cover was even 6.1 °C lower than that of ambient on average, signifying possible passive cooling without consuming any energy when using the ACA panel as building envelope.

Fig. 5 Cooling performance of the large-scale ACA panel. (a) Thermal conductivity and solar weighted reflectance of the ACA panel at different regions, showing a highly consistent performance. (b) Solar irradiance (G) and internal temperature (T) changes in different samples in the outdoor tests. (c) Selective absorptivity/emissivity spectra of the ACA in the solar and LWIR wavelengths. (d) Year-round cooling energy consumption of a building model with and without an ACA envelope and the potential cooling energy savings by using ACA in 20 different cities. (e) Year-round energy savings in different climate regions in China.”

Supplementary Information:

“

Supplementary Fig. 17. (a) Digital photograph and (b) schematic showing the set-up used for outdoor tests. All thermocouples were covered by aluminum foil to prevent exposure to sunlight.”

4. Fig. 5e: Why the fluctuation of the glass temperature was much more significant than that of the other three panels? Is this because the thermocouple probe attached to the glass is exposed to the sunlight directly? Typically, the thermocouple probe should be covered by aluminum foil to prevent exposure to the sunlight.

Response:

When we redid the outdoor tests, all thermocouples were covered by aluminum foil to prevent exposure to sunlight, as shown in Supplementary Fig. 17b. Nevertheless, the large fluctuation of temperature was still observed in the glass sample, as shown in Fig. 5b. It is also noted that the trend of fluctuation closely followed that of the solar irradiance. We suspect it is likely due to the high thermal conductivity ($0.8 \text{ W m}^{-1} \text{ K}^{-1}$) and transparency of glass, making the interior temperature under glass highly dependent on solar irradiance. A similar dependence of temperature fluctuation on solar irradiance was also reported in a previous study (*ACS Appl Mater Interfaces* 2020, **12**, 51409–51417).

The above discussion on the temperature fluctuation is included in the revision.

Page 14:

“The interior temperature under the glass cover fluctuated following the trend of solar irradiance likely due to its high thermal conductivity ($0.8 \text{ W m}^{-1} \text{ K}^{-1}$) and transparency, allowing direct transmission of sunlight into the interior.⁷⁰ By contrast, the other opaque samples showed relatively stable internal temperatures which were much lower than that of transparent glass.”

References:

“70. Zhong H, Zhang P, Li Y, Yang X, Zhao Y, Wang Z. Highly solar-reflective structures for daytime radiative cooling under high humidity. *ACS Appl Mater Interfaces* **12**, 51409-51417 (2020).”

5. Lines 415-416, Page 17: Why use a PE film atop the ACA panel? The panel itself should be the exterior surface of the building envelope in the real-world application (just as the authors illustrated in Fig. 1a).

Response:

The purpose of using a PE film was to reduce the convective heat loss, consistent with previous works (*Nat Nanotechnol* 2021, **16**, 153–158; *Joule* 2019, **3**, 111–123.). As suggested by the reviewer, we modified the setup for outdoor testing by exposing the test samples as the exterior surface to reflect real-world applications (Supplementary Fig. 17). A PE film was placed underneath the sample to minimize the convection between the interior and ambient environment.

The above discussion is included in the revision:

Page 19:

“During the outdoor tests, the exterior surfaces of the test samples were directly exposed to the sunlight, simulating the scenario when they were used as the building envelope. A polyethylene film was placed beneath the test samples to minimize the convection between the interior and ambient environment (Supplementary Fig. 17).^{80, 81}”

References:

“80. Li D, et al. Scalable and hierarchically designed polymer film as a selective thermal emitter for high-performance all-day radiative cooling. *Nat Nanotechnol* **16**, 153-158 (2021).

81. Zhao D, et al. Subambient cooling of water: toward real-world applications of daytime radiative cooling. *Joule* **3**, 111-123 (2019).”

6. Besides, calibration of the thermocouples is needed. Error analysis of the outdoor testing results should also be included.

Response:

As suggested, we have included the calibration of thermocouples and the error estimation of the outdoor testing in the revision:

Page 20:

“The thermocouples were calibrated by CENTER Technology using the standard traceable to National Institute of Standards and Technology. The measurement errors for the outdoor tests were estimated by measuring the interior temperatures of three different ACA samples simultaneously under the same condition (Supplementary Fig. 18). They exhibited quite consistent variations in internal temperatures with an average standard deviation of ± 0.3 °C and a maximum standard deviation not exceeding ± 1.0 °C, indicating acceptable reliability of outdoor test results.”

Supplementary Fig. 18:

“

Supplementary Fig. 18. (a) Solar irradiance, G , and internal temperatures of three ACA samples measured in the outdoor tests. (b) Average temperatures of three samples and the standard deviations.”

7. A simple comparison of stagnation temperatures among the four selected materials is inadequate to demonstrate the energy-saving potential of the ACA material. Building energy simulation is a necessity to quantitatively predict the year-round potential impact of the ACA on building energy efficiency in different climate regions.

Response:

As suggested, the potential of ACA for year-round building energy saving in different climate regions was evaluated quantitatively using EnergyPlus (version 9.6.0) in the revision:

Page 15:

“The energy-saving potential of ACA in different climate regions was demonstrated by comparing the year-round cooling energy consumption of a building model with and without an ACA envelope using EnergyPlus (version 9.6.0) (see calculation details in Section S6, Supplementary Information). As shown in Fig. 5d, the cooling energy consumptions for all 20 cities across China were significantly reduced when ACA was used, achieving a 56.0% cooling energy saving on average compared to the baseline without ACA. The year-round savings of cooling energy for different climate regions in China are highlighted in Fig. 5e. The application of ACA in tropical and subtropical climates had greater potential for cooling energy savings with more than 10 MJ m^{-2} annual saving in southern China than in cooler temperate regions like northern China.

Fig. 5 Cooling performance of the large-scale ACA panel. (a) Thermal conductivity and solar weighted reflectance of the ACA panel at different regions, showing a highly consistent performance. (b) Solar irradiance (G) and internal temperature (T) changes in different samples measured in outdoor tests. (c) Selective absorptivity/emissivity spectra of the ACA in the solar and LWIR wavelengths. (d) Year-round cooling energy consumption of a building model with and without an ACA envelope and the potential cooling energy savings by using ACA in 20 different cities. (e) Year-round energy savings in different climate regions in China.”

Supplementary Information:

“S6. Cooling energy saving simulation

To evaluate the potential of ACA for cooling energy saving, a building energy simulation was performed using an open-source software EnergyPlus version 9.6.0. The building model without ACA (Supplementary Fig. 19) consisted of envelopes with baseline properties, including roofs (with an emissivity of 0.9, solar reflectivity of 0.3 and thermal conductivity of $0.16 \text{ W m}^{-1} \text{ K}^{-1}$) and walls (with an emissivity of 0.9, solar reflectivity of 0.1 and thermal conductivity of $0.6 \text{ W m}^{-1} \text{ K}^{-1}$) according to previous studies.^{9, 10} For the building model with ACA, an ACA envelope with thermo-optical properties obtained in this work was placed on top of the baseline envelope. The indoor temperature was controlled at $26 \text{ }^{\circ}\text{C}$ by an HVAC system, which is an economic temperature recommended by the U.S. Department of Energy.¹¹ The cooling energy consumptions of the building models with and without ACA were calculated for different cities in China based on the climate data downloaded from the EnergyPlus website (<https://energyplus.net/weather>).

Supplementary Fig. 19. Schematic of a simplified residential house with floor area of 927 m² used for EnergyPlus simulation.”

Supplementary References:

- “9. Zhong H, et al. Hierarchically Hollow Microfibers as a Scalable and Effective Thermal Insulating Cooler for Buildings. *ACS Nano* **15**, 10076-10083 (2021).
10. Cai C, et al. Dynamically Tunable All-Weather Daytime Cellulose Aerogel Radiative Supercooler for Energy-Saving Building. *Nano Lett* **22**, 4106-4114 (2022).
11. Lin C, et al. All-weather thermochromic windows for synchronous solar and thermal radiation regulation. *Sci Adv* **8**, eabn7359 (2022).”

8. Overall, the proposed scheme, namely, developing an aerogel-based building envelope material with ultralow thermal conductivity and high solar reflectance for building energy-saving, is not novel enough. The performance of the large-scale ACA material is also not sufficiently good. Besides, one can easily achieve the goal proposed in this work by combining an aerogel with even lower thermal conductivity and a top layer of radiative cooling coating with higher solar reflectance and much better weather resistance.

Response:

Contrary to the reviewer’s comment, we believe the novelty of this work is multifold beyond the development of aerogel-based building envelope having an ultralow thermal conductivity and a high solar-reflectance.

First, a novel additive freeze-casting technique was for the first time developed to fabricate large-scale anisotropic aerogel panels with horizontally uniform pore alignment from the bottom-up approach, which could only be achieved previously by a top-down method applicable to few selected raw materials such as cellulose (*Sci Adv* 2018, **4**, eaar3724). The bottom-up approach developed in this work can be extended to other nanomaterials and polymer systems, expanding the types of large-scale aerogels with uniformly aligned in-plane pores. To highlight the vital role played by additive freeze-casting in achieving aligned pores for thermal insulation, we further compared the insulation performance of two aerogel samples made by additive freeze-casting and traditional one-step freeze-casting in the revision, as shown in Supplementary Fig. 16. The lower thermal conductivity than air was only attained in

the additive freeze-cast aerogel thanks to the consistent pore alignment, whereas a higher thermal conductivity than air was observed in the aerogel made by conventional freeze-casting.

Second, the uniformly aligned BNNS in the pore walls created by additive freeze-casting translated the anisotropic k of BNNS into the ACA, leading to an ultralow k in the thickness direction for effective thermal insulation.

Third, the optical properties of BNNS have been mostly overlooked previously for thermal management applications. In comparison, our ACA leveraged the high refractive index of BNNS to reduce the solar heat gain, delivering an excellent solar reflectance together with an ultralow k . The last two characteristics of additive freeze-cast ACA were made possible by exploiting the unique thermo-optical properties of BNNS, distinctively different from previous thermal management solutions with BNNS where only the high in-plane k of BNNS was utilized (*Adv Funct Mater* 2021, **31**, 2008705; *ACS Nano* 2018, **13**, 337-345).

In terms of the performance of our ACA, we have included a comprehensive comparison with different thermal insulation and radiative cooling materials reported in the literature, as shown in Supplementary Table 3. First, the thermal conductivity of our ACA ($16.9 \text{ mW m}^{-1} \text{ K}^{-1}$ in the transverse direction) was among the lowest values ever achieved by thermal insulation materials. There are only a few thermal insulation materials exhibiting lower thermal conductivities than our ACA, including GO/nanocellulose foam ($15 \text{ mW m}^{-1} \text{ K}^{-1}$ from *Nat Nanotechnol* 2015, **10**, 277-283), closed-cell graphene foam ($5.75 \text{ mW m}^{-1} \text{ K}^{-1}$ from *Adv Funct Mater* 2021, **31**, 2007392), cellulose aerogel ($15.5 \text{ mW m}^{-1} \text{ K}^{-1}$ from *ACS Appl Mater Interfaces* 2020, **12**, 45363-45372), silica/nanocellulose aerogels ($13.8 - 20.1 \text{ mW m}^{-1} \text{ K}^{-1}$ from *Adv Funct Mater* 2015, **25**, 2326-2334), and hollow microfiber film ($14 \text{ mW m}^{-1} \text{ K}^{-1}$ from *ACS Nano* 2021, **15**, 10076-10083). However, most of these insulating materials failed to show a high solar reflectance except for the hollow microfiber film. The current ACA delivered a higher solar reflectance (97%) than the hollow microfiber film (94%), making it a superior choice than any other insulation materials for building envelopes. Second, the solar reflectance of our ACA (97%) is among the best even compared to those radiative cooling coatings or films, whose reflectance ranged from 93% to 98%. Third, the energy-saving potential of ACA in different climate regions was also demonstrated using EnergyPlus simulation (Fig. 5e), showing a 56.0% cooling energy saving on average across 20 different cities in China compared to that of baseline without ACA. All in all, the above analysis signifies extraordinary overall performance of the current ACA, better than existing thermal insulation and radiative cooling materials.

The above comparison also indicates that the solar reflection of ACA is among the highest radiative cooling materials while possessing the thermal conductivity among the lowest thermal insulation materials. Therefore, the best thermal insulation material and the best radiative cooling material are likely needed to beat the overall performance of the current ACA if a combination of two materials were to be used as suggested by the reviewer. While we were not able to identify any such combination in the literature, we envisioned that the cost, the interfaces between the two materials, as well as the scalability could be potential issues for their practical application. By contrast, our ACA achieved both desired properties in the same material and demonstrated its scalability with decimeter-long panels. Moreover, the weather resistance of ACA is now validated in the revision (see our response to Comment #2). Based on the above, we believe our ACA is a better choice of material than the bilayer structure containing an aerogel with a radiative cooling coating.

The above discussion is now incorporated in the revision to further highlight the novelty and performance of ACA.

Page 14:

“It should be noted that the lower k_{trans} values than that of air were attributed to the consistent pore alignment of ACA thanks to the uniform ice crystal growths facilitated by additive freeze-casting (Supplementary Fig. 16a). By contrast, conventional one-step freeze-casting led to random pore alignments when the freezing front advanced away from the fixed cold source (Supplementary Fig. 16b), resulting in a high k_{trans} of $28.5 \pm 2.8 \text{ mW m}^{-1} \text{ K}^{-1}$ exceeding that of the air (Supplementary Fig. 16c). Moreover, the better alignment achieved by additive freeze-casting also offered a higher k_{align} value of ACA than that made by conventional one-step freeze-casting, giving rise to a higher anisotropic factor (R) of the former for a superior thermal insulation performance.”

Supplementary Fig. 16:

“

Supplementary Fig. 16. Schematics showing the ice crystal growths in (a) additive freeze-casting and (b) conventional one-step unidirectional freeze-casting. (c) Comparison of thermal conductivities and anisotropic factors of two aerogels made by different freeze-casting methods.”

Page 16:

“The incorporation of 2D BNNS offered two distinct characteristics to the additive freeze-cast ACA compared to previous thermal management solutions. First, although BNNS has been extensively used for cooling applications, most efforts were focused on utilizing the high in-plane k of BNNS to dissipate the heat, e.g., for personal⁷⁸ and electronics cooling,⁷⁹ while the anisotropic k of BNNS was not fully exploited. Here, the consistently aligned BNNS in the pore walls by additive freeze-casting translated the anisotropic k of BNNS into the ACA, leading to an ultralow k in the thickness direction for effective thermal insulation. Second, the optical properties of BNNS have been mostly overlooked previously for thermal management applications. By contrast, our ACA leveraged the high refractive index of BNNS to reduce the solar heat gain, delivering an excellent solar reflectance together with an ultralow k .”

Page 17:

“The solar reflection of ACA was among the highest radiative cooling materials while the thermal conductivity was among the lowest thermal insulation materials (Supplementary Table 3). The excellent dual functionalities enabled the ACA to lessen both parasitic and solar heat gains when used as cooling panel under direct sunlight, potentially superior to a combination of thermal insulation aerogel and radiative cooling coating to significantly reduce the energy consumption for cooling applications.”

References:

78. Miao D, Wang X, Yu J, Ding B. A biomimetic transpiration textile for highly efficient personal drying and cooling. *Adv Funct Mater* **31**, 2008705 (2021).

79. Chen J, Huang X, Sun B, Jiang P. Highly thermally conductive yet electrically insulating polymer/boron nitride nanosheets nanocomposite films for improved thermal management capability. *ACS Nano* **13**, 337-345 (2018).”

Supplementary Table 3:

“S8. Comparison of overall performance of ACA with other materials

Supplementary Table 3. Thermal insulation and solar reflectance performance of ACA compared to other thermal insulation materials and high-reflectance or radiative cooling coatings.

Materials	Thermal conductivity (mW m ⁻¹ K ⁻¹)	Solar reflectance (%)	Contact angle
Thermal insulation materials			
Cotton stalk fibers ¹³	58.5 – 81.5	/	/
Durian peel and coconut coir ¹⁴	134.2	/	/
Vermiculite, sunflower and wheat stalk ¹⁵	63 – 334	/	/
Corn Stalk ¹⁶	51	/	/
Wood waste ¹⁷	48 – 55	/	/
PVA fiber aerogel ¹⁸	319.7	/	/
Coal fly ash composite foam ¹⁹	51.1	/	/
Polyurethane foam ²⁰	20 – 30	/	/
Expanded polystyrene foam ²⁰	30 – 40	/	/
Extruded polystyrene foam ²¹	25 – 35	/	/
Glass wool ²¹	30 – 46	/	/
Rock wool ²¹	33 – 46	/	/
Liquid-crystalline nanocellulose aerogel ²²	18	/	/
GO/nanocellulose foam ²³	15 (transverse) 170 (alignment)	/	/
CNF/ZrP/RGO aerogel ²⁴	18 (transverse) 45 (alignment)	/	/
PVA/CNF aerogel ²⁵	38.0	/	/
CNF/MoS ₂ aerogel ²⁶	28.09	/	Hydrophilic
Attapulgit/gelatin composite aerogel ²⁷	34.28 – 35.29	/	/
ZIFs/PVA aerogel ²⁸	32.7 – 36.3	/	/
HAP/PVA aerogel ²⁹	33.6 – 38.7	/	150°
CNF aerogel ³⁰	25.5	/	/

SBC/CNF aerogel ³¹	28	/	/
PI-BNC aerogel ³²	23 (transverse)	/	/
	44 (alignment)		
BCF-CNF aerogel ³³	23	/	/
Cellulose nanofibril/emulsion aerogel ³⁴	15.5	/	/
Cotton NFCs ³⁵	39.6 – 45.5	/	/
SiO ₂ -CNFs ³⁶	13.8 – 20.1	/	Hydrophobic
Silica granular aerogels ³⁷	24	/	/
BNNS/PVA aerogels ³⁸	23.5	93.8	/
Hypocrystalline ceramic aerogels ³⁹	26	/	/
PE aerogel ⁴⁰	28	92.2	/
PDMS/PE aerogel ⁴¹	32	96	155°
Superhydrophobic cellulose aerogel ⁴²	28	93	152°
CNC aerogels ¹⁰	26	96	138°
Hollow microfibers cooler ⁹	14	94	Hydrophobic
Graphene-based foam ⁴⁴	5.75	/	/
Bioinspired polymeric woods ⁴	20.8	/	120° – 150°
Nanowood ⁴⁵	30 (transverse)	95	/
	60 (alignment)	(0.4 – 1.1 μm)	
High-reflectance/radiative cooling coatings and films			
Silica aerogel/polyurethane film ⁴⁶	/	69 – 89	101° – 135°
Nanoporous polymer film ⁴⁷	/	96.2	/
Porous P(VdF-HFP)HP coatings ⁴⁸	/	96	110°
BaSO ₄ nanocomposite paints ⁴⁹	/	98.1	Water resistant
TiO ₂ /Silica aerogel nanocomposite paint ⁵⁰	29	90	142°
ePTFE film/Ag layer ⁵¹	/	98	142°
PMMA film ⁵²	/	95	156°
Electro-spun PAN nanofibers film ⁵³	/	95	/
Nanoporous composite fabric (NCF) ⁵⁴	/	95	109° after 20 min
SiO ₂ /EPDM porous composite film ⁵⁵	/	96	162°
TiO ₂ -free coatings ⁵⁶	/	96	138.9°
Superhydrophobic PTFE coating ⁵⁷	/	93	165°
TiO ₂ /glass microsphere/polymer emulsion ⁵⁸	/	93.4	109.4°
ACA (This work)	16.9 (transverse)	97	110°
	98.4 (alignment)		

»

REVIEWERS' COMMENTS

Reviewer #3 (Remarks to the Author):

The authors addressed all my concerns and significantly improved the paper. It is acceptable for publication.

Reviewer #4 (Remarks to the Author):

The authors have addressed most of my comments properly. I have several minor concerns:

1. It seems that all the questions and comments from Reviewer 2 are not addressed in the response letter. Please check.
2. For question 2, the authors have proven that the ACA has good weather resistance. Specifically, hydrophobicity of the ACA is demonstrated. My concern is that if the longitudinal section (not the top surface) of the ACA is covered by water, will the hydrophobicity still exist? Will the water be transferred along the pore direction?
3. For outdoor testing, a glass case is used as a reference. It should be noted that glass is transparent for sunlight, so the temperature of the inner environment will be affected directed by the transmitted sunlight. Thus, it is unfair for the glass case during the comparison process. This point should be clarified in the manuscript.
4. Small fraction of the manuscript is prepared for spectrum design and optimization. Whether the structure optimization (size, porosity, and so on) for the spectrum and thermal conductivity is synergetic? If possible, give a discussion and description.

Authors' response to Reviewers' Comments:

We would like to thank the reviewers for careful reading of our revised manuscript (NCOMMS-22-10974A) as well as their additional comments and constructive suggestions, which were tremendously helpful in further improving the quality of this manuscript. We have addressed all the comments by the reviewers and revised our manuscript according to their suggestions. The amendments made to individual comments are summarized below. We hope that the revision is now acceptable for publication in *Nature Communications*.

Reviewer #3

The authors addressed all my concerns and significantly improved the paper. It is acceptable for publication.

Response: We thank the reviewer for the constructive comments.

Reviewer #4

The authors have addressed most of my comments properly. I have several minor concerns:

1. It seems that all the questions and comments from Reviewer 2 are not addressed in the response letter. Please check.

Response: We have revised the manuscript accordingly. Given the editor's consideration of Reviewer #2's comments not being particularly relevant to this work, we have not specifically addressed them item by item. However, we have seriously taken into account important ones in a holistic manner in our first revision. For example, (i) the summary of the central core of knowledge and the focus of the paper in abstract, (ii) the addition of detailed description of the critical points

and trends in figures, and (iii) the discussion of major findings in greater depth, (iv) the chronological description of research methods, and so on, were incorporated.

2. For question 2, the authors have proven that the ACA has good weather resistance. Specifically, hydrophobicity of the ACA is demonstrated. My concern is that if the longitudinal section (not the top surface) of the ACA is covered by water, will the hydrophobicity still exist? Will the water be transferred along the pore direction?

Response: Because of the highly porous cross-section, we expect that water droplets would gradually penetrate the sample through the pores when the lateral surface is exposed. Similar to the top surface, we applied the waterproof treatment on the lateral surface of ACA, as shown in Supplementary Fig. 21e. Given the lateral surface not being directly exposed to the sunlight, it is envisaged that the waterproof coating would not affect the overall reflectance and thermal conduction performance of the ACA.

The above discussion has been included in the revision.

Page 16:

“The lateral surfaces of ACA were highly porous and thus prone to water uptake when exposed. The waterproof PU coating was also applied on the lateral surfaces to enhance their hydrophobicity and avoid the water transfer through the longitudinal pore channels (Supplementary Fig. 21e).”

Supplementary Fig. 21. ... (e) Photographs showing the changes of water droplets on the PU-coated lateral surface of ACA. Scale bars in (d) and (e): 2 cm.

3. For outdoor testing, a glass case is used as a reference. It should be noted that glass is transparent for sunlight, so the temperature of the inner environment will be affected directed by the transmitted sunlight. Thus, it is unfair for the glass case during the comparison process. This point should be clarified in the manuscript.

Response: As suggested, we have clarified the use of glass and removed the comparison between ACA and glass in the revision.

Page 15:

“It should be noted that the interior temperature under the glass panel was directly affected by the transmitted sunlight. Therefore, we kept the comparison of internal temperature to opaque materials. Overall, our ACA panel maintained the lowest interior temperature throughout the whole period among all opaque materials studied, as shown in Fig. 5b. Between 14:00 and 15:00, when the solar irradiance was $640 - 900 \text{ W m}^{-2}$ (Fig. 5b), our ACA panel maintained an average

temperature of 35.0 °C, which was 7, 11, and 13 °C lower than the SiO₂ aerogel, EPS foam, and brick counterparts, respectively.”

4. Small fraction of the manuscript is prepared for spectrum design and optimization. Whether the structure optimization (size, porosity, and so on) for the spectrum and thermal conductivity is synergetic? If possible, give a discussion and description.

Response: In the original manuscript, we discussed the effect of WPU concentration on the porosity and the reflectance spectra, as shown in Supplementary Fig. 11. The decreasing WPU concentration led to a rising porosity, translating into an increasing reflectance over the whole solar spectrum. The optimal WPU concentration of 1.4 wt% for high solar reflectance also delivered the lowest thermal conductivity, as shown in Supplementary Fig. 8, indicating potential synergy arising from the optimal WPU concentration to deliver both desired solar reflectance and thermal conductivity. As suggested, we have included further discussion on the structural optimization for balanced solar reflectance and thermal conductivity through controlling the BNNS loading in the revision.

Page 12:

“The solar reflectance increased from 8 to 94 % when the WPU structure was changed from a solid film (with a 0% porosity) to an aerogel at a WPU concentration of 1.4 wt% (with porosity of 97.7%) (Supplementary Fig. 11). The same aerogel also delivered the lowest k_{trans} value (Supplementary Fig. 8), indicating potential synergy arising from the optimal WPU concentration for desired solar reflectance and thermal conductivity.”

“The experimentally measured solar-weighted reflectance initially surged to 97% when the BNNS loading was increased to 10 wt %, followed by a gradual decline with further increasing the BNNS loading (Supplementary Fig. 13a). Of note is that such an optimal BNNS loading for the high reflectance also led to the lowest thermal conductivity in the transverse direction (Fig. 3a), both of which resulted from the highest porosity achieved at a BNNS loading of 10 wt% (Fig. 2c).”

Supplementary Fig. 13:

“

Supplementary Fig. 13. (a) Solar-weighted reflectance of 1.4 wt% WPU aerogels with different BNNS loadings. (b) Solar reflection spectra of 1.4 wt% WPU aerogels with and without BNNS.”